# RNA-dependent chromatin association of transcription elongation factors and Pol II CTD kinases

**Sofia Battaglia[1†], Michael Lidschreiber[1,2†], Carlo Baejen[1], Phillipp Torkler[1], Seychelle M Vos[1], Patrick Cramer[1,2]***

[1]Department of Molecular Biology, Max Planck Institute for Biophysical Chemistry, Göttingen, Germany; [2]Department of Biosciences and Nutrition, Center for Innovative Medicine and Science for Life Laboratory, Novum, Karolinska Institutet, Huddinge, Sweden

**Abstract** For transcription through chromatin, RNA polymerase (Pol) II associates with elongation factors (EFs). Here we show that many EFs crosslink to RNA emerging from transcribing Pol II in the yeast *Saccharomyces cerevisiae*. Most EFs crosslink preferentially to mRNAs, rather than unstable non-coding RNAs. RNA contributes to chromatin association of many EFs, including the Pol II serine 2 kinases Ctk1 and Bur1 and the histone H3 methyltransferases Set1 and Set2. The Ctk1 kinase complex binds RNA in vitro, consistent with direct EF-RNA interaction. Set1 recruitment to genes in vivo depends on its RNA recognition motifs (RRMs). These results strongly suggest that nascent RNA contributes to EF recruitment to transcribing Pol II. We propose that EF-RNA interactions facilitate assembly of the elongation complex on transcribed genes when RNA emerges from Pol II, and that loss of EF-RNA interactions upon RNA cleavage at the polyadenylation site triggers disassembly of the elongation complex.

*For correspondence: patrick. cramer@mpibpc.mpg.de

†These authors contributed equally to this work

Competing interests: The authors declare that no competing interests exist.

## Introduction

For productive transcription through chromatin, RNA polymerase (Pol) II associates with general elongation factors (EFs) (*Perales and Bentley, 2009*; *Shilatifard, 2004*; *Shilatifard et al., 2003*; *Sims et al., 2004*) that are recruited to the body of transcribed genes in yeast (*Mayer et al., 2010*). EFs in yeast include Spt5 (a subunit of human DSIF), the histone chaperone Spt6, and the Paf1 complex (Paf1C). The Pol II C-terminal domain (CTD) kinases Bur1 (human CDK9) and Ctk1 (human CDK12), and their cyclin partners Bur2 and Ctk2, respectively, can also be classified as EFs. In addition, the histone methyltransferases Set1 (a subunit of the COMPASS complex), Set2, and Dot1, are recruited to elongating Pol II to set the 'active' histone marks H3K4me3, H3K36me3, and H3K79me3, respectively.

Despite extensive research, it remains unclear for several EFs how they are recruited to active genes. EFs may be recruited by interactions with the body of transcribing Pol II, or by contacts with the tail-like C-terminal repeat domain (CTD) of Pol II, or they may bind via other Pol II-associated EFs. Spt5 binds the body of the Pol II elongation complex (*Grohmann et al., 2011*; *Klein et al., 2011*; *Martinez-Rucobo et al., 2011*), whereas Bur1, Spt6 and Set2 bind the CTD (*Dengl et al., 2009*; *Kizer et al., 2005*; *Li et al., 2003*; *Phatnani et al., 2004*; *Sun et al., 2010*; *Yoh et al., 2007*; *Qiu et al., 2009*; *Li et al., 2002*). Interaction of Paf1C with Pol II involves Spt5 (*Liu et al., 2009*; *Mayekar et al., 2013*; *Wier et al., 2013*; *Zhou et al., 2009*; *Qiu et al., 2012, 2009*) and the CTD (*Qiu et al., 2012*), whereas interaction of Set1 with Pol II involves Paf1C (*Krogan et al., 2003a*; *Ng et al., 2003*).

However, other recruitment mechanisms exist because mutations in EFs that prevent their polymerase interactions do not abolish gene occupancy of such factors, including Bur1, Paf1C subunits, Spt6, and Set2 (*Ng et al., 2003*; *Qiu et al., 2012*, *2009*; *Mayer et al., 2010*; *Zhou et al., 2009*; *Krogan et al., 2003b*). Further, it remains unknown how the yeast CTD serine 2 (Ser2) kinase Ctk1 is recruited, which is apparently a prerequisite for recruitment of Spt6 and Set2, because these factors bind the Ser2-phosphorylated CTD (*Dengl et al., 2009*; *Kizer et al., 2005*; *Li et al., 2003*; *Phatnani et al., 2004*; *Sun et al., 2010*; *Yoh et al., 2007*). More generally, it is unknown whether and how EFs can distinguish transcribing Pol II from free or initiating polymerase based on polymerase interactions alone, in particular at an early stage of elongation when Ser2 phosphorylation is absent.

An alternative mechanism of EF recruitment would involve interactions with the nascent pre-mRNA. Such RNA interactions are well established for RNA processing factors that are recruited during Pol II elongation (*Perales and Bentley, 2009*; *Bentley, 2005*; *Baejen et al., 2014*; *Tuck and Tollervey, 2013*) for co-transcriptional capping (*Martinez-Rucobo et al., 2015*), splicing (*Bentley, 2005*; *Saldi et al., 2016*), and 3′-processing (*Proudfoot, 2011*; *Shi and Manley, 2015*) of the pre-mRNA. Some observations indeed suggest that nascent RNA contributes to the recruitment of EFs to Pol II. Spt5 and Set1 bind RNA in vitro (*Meyer et al., 2015*; *Missra and Gilmour, 2010*; *Trésaugues et al., 2006*; *Halbach et al., 2009*), Ctk1 and Bur1 in vivo occupancy at active genes depends on the cap-binding complex, which binds 5′-capped RNA (*Hossain et al., 2013*; *Lidschreiber et al., 2013*), and Paf1C binds RNA, which is required for full gene occupancy (*Dermody and Buratowski, 2010*).

Here we report that most EFs in yeast, including, most notably, CTD Ser2 kinases and histone H3 methyltransferases, directly crosslink to nascent pre-mRNA in vivo. We find that crosslinking preferences can differ for coding RNAs and non-coding (nc) RNAs. We further show that chromatin association of many EFs depends on RNA. We also directly tested one prominent EF for RNA binding in vitro, and found that recombinant, purified Ctk1-containing kinase complex CTDK-I strongly binds RNA in the absence of other components. Moreover, we show that the N-terminal region of Set1 that contains two RNA recognition motifs (RRMs) is required for full Set1 recruitment to genes in vivo. Based on these results we suggest a model where nascent RNA contributes to the assembly and stability of the Pol II elongation complex. RNA-EF interactions provide a missing link for understanding the coordination of the transcription cycle.

## Results

### Elongation factors directly crosslink to RNA in vivo

To investigate whether EFs interact with RNA in vivo, we used photoactivatable ribonucleoside-enhanced crosslinking and immunoprecipitation (PAR-CLIP) (*Hafner et al., 2010*), a method that detects and maps direct protein-RNA interactions without chemical crosslinkers. We applied our recently optimized PAR-CLIP protocol (*Baejen et al., 2014*) to 14 EFs of the yeast *Saccharomyces cerevisiae* (*Table 1*, Materials and methods). These EFs included Spt5, Spt6, the five Paf1C subunits Cdc73, Ctr9, Leo1, Rtf1, and Paf1, the kinases Bur1 and Ctk1, the cyclins Bur2 and Ctk2, and the histone methyltransferases Set1, Set2, and Dot1.

For 12 of these 14 EFs we obtained PAR-CLIP signals that were more than two-fold above background, showing that these EFs interact with RNA in vivo (*Figure 1*, *Figure 1—figure supplement 1A*). We obtained between 42,000 and 520,000 high-confidence protein-RNA crosslinking sites per factor with p-values below 0.005 (*Table 1*). The obtained data sets were highly reproducible (*Figure 1—figure supplement 1B*). To estimate background RNA binding, we collected PAR-CLIP data for the transcription initiation factor TFIIB that is recruited to promoter DNA before nascent RNA is made (*Sainsbury et al., 2015*). Only very low levels of background binding were observed, further emphasizing the significance of EF-RNA interactions detected by UV crosslinking.

We then classified EFs into factors with moderate and high PAR-CLIP signals, based on their fold enrichments (>2 and >4 fold, respectively) over background TFIIB signals (*Figure 1*). Spt5, Set1, Ctk1, Spt6, Ctk2 and Bur1 showed high PAR-CLIP signals (*Figure 1*, *Figure 1—figure supplement 1A*, *Table 1*). EFs with moderate signals included Rtf1, Ctr9, Cdc73, Bur2, Set2 and Dot1. PAR-CLIP signals were clearly specific for individual subunits of known complexes. For instance, only the Paf1C

**Table 1.** PAR-CLIP analysis of elongation factors (EFs).

| EF | Complex* | Number of crosslink sites[†] |
|---|---|---|
| Bur1 | BUR kinase complex | 77931 |
| Bur2 | | 46293 |
| Ctk1 | CTDK-I | 129352 |
| Ctk2 | | 98993 |
| Cdc73 | Paf1C | 57603 |
| Ctr9 | | 55807 |
| Leo1 | | 27665 |
| Paf1 | | 20742 |
| Rtf1 | | 60068 |
| Set1 | COMPASS | 189723 |
| Set2 | | 68875 |
| Dot1 | | 42848 |
| Spt5[‡] | DSIF | 517568 |
| Spt6 | | 93902 |
| TFIIB[§] | | 16686 |

*DSIF, DRB sensitivity inducing factor; CTDK, C-terminal domain kinase; Paf1C, Paf1 complex; COMPASS, Complex Proteins Associated with Set1.

[†]Average number of crosslink sites with p-values<0.005.

[‡](**Baejen et al., 2017**).

[§]Initiation factor, used to determine the level of RNA background crosslinking

subunits Rtf1, Cdc73 and Ctr9 bound RNA according to the PAR-CLIP results, and the same subunits bound radioactively labeled RNA after immunoprecipitation (*Figure 1—figure supplement 1C*). A very low background signal was observed for other subunits, whereas the enriched bands were due to the protein of interest. These data revealed that many EFs directly bind RNA in vivo, including Pol II Ser2 kinases and histone H3 methyltransferases.

## Comparisons of PAR-CLIP data require normalization

We have previously noted the importance of normalizing the raw PAR-CLIP signal, as measured by the number of U-to-C transitions per U site, to account for differences in RNA abundance (*Baejen et al., 2014*). Briefly, the raw PAR-CLIP signal is proportional to the occupancy of the factor on RNA and to the concentration of RNAs covering the U site. Therefore, normalization is crucial to enable comparison of PAR-CLIP signals between individual transcripts and transcript classes. Relative occupancies can be estimated by dividing the observed PAR-CLIP signal by RNA-Seq reads that have been obtained under the same experimental conditions (*Baejen et al., 2014*). An alternative approach is to divide the observed PAR-CLIP signal by a PAR-CLIP signal obtained for Pol II (*Baejen et al., 2017*), although this is only suitable for proteins that associate with nascent RNA during transcription, which is the case for the EFs studied here.

In *Figure 2* we investigate how the two different normalization methods affect EF occupancy profiles on mRNA transcripts. For two representative EFs, Ctk2 and Spt5, the raw data (*Figure 2A*) was either normalized with RNA-Seq reads (*Figure 2B*) or with reads from Pol II (Rpb1 subunit) PAR-CLIP data (*Figure 2C*). Meta-transcript profiles are shown in *Figure 2D*. In the case of Ctk2, the raw data profile and the Pol II normalized profile look very similar, whereas the RNA-normalized profile shows slightly less occupancy of Ctk2 in the 3′ part of the transcripts, due to the slightly higher RNA-Seq signal in this region (*Figure 2B*, bottom). The PAR-CLIP signal for Spt5 is enriched around the 5′-end of mRNAs, decreases towards the 3′-end, and this was independent of the normalization approach (*Figure 2D*, bottom). However, Spt5 signals peak just downstream of the pA site, and the size of this peak varies dependent on the normalization approach. This is due to the intrinsic instability of

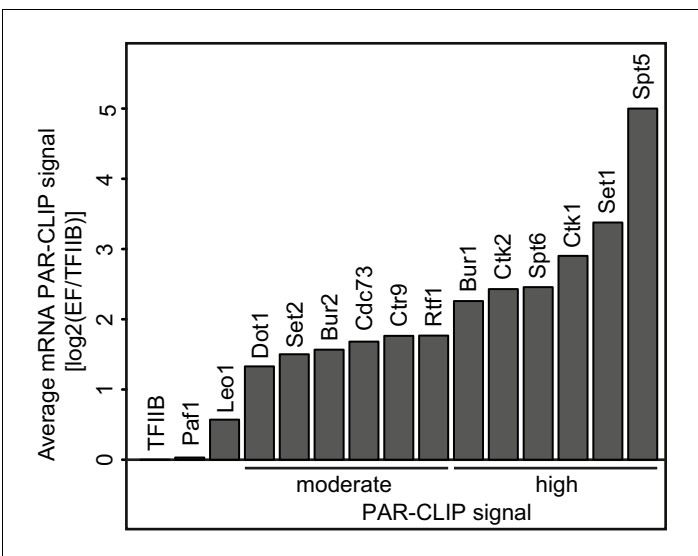

**Figure 1.** Many elongation factors (EFs) bind RNA in vivo. PAR-CLIP signal strength for EFs varies. The barplots show log2 fold-enrichments of transcript-averaged PAR-CLIP signals over the averaged PAR-CLIP signal for initiation factor TFIIB, which shows background RNA binding. Averaged PAR-CLIP signals were calculated by taking mean transcript PAR-CLIP signals averaged over all mRNAs, which were filtered to be 800–5000 nt long and at least 150 nt away from neighboring transcripts (2532 mRNAs). Heat plots averaged over mRNA transcripts of the corresponding PAR-CLIP signals are shown in *Figure 1—figure supplement 1A*.

The following figure supplement is available for figure 1:

**Figure supplement 1.** Confirmatory information on PAR-CLIP experiments.

transcripts downstream of the pA site, which reduces the number of RNA-Seq reads, and artificially increases the PAR-CLIP peak after RNA-Seq-based normalization.

Taken together, the PAR-CLIP metagene profiles over stable transcripts were largely independent of the type of normalization used, whereas normalization becomes very important when crosslinking to unstable RNAs is investigated. Indeed, when we compare meta-profiles over cryptic unstable transcripts (CUTs) versus stable mRNAs using the different normalization methods (*Figure 2—figure supplement 1*), we observe that for proteins that bind CUTs (e.g. Spt5) the relative signal over CUTs increases when total RNA-Seq reads are used for normalization, similarly as for unstable transcripts downstream of the pA site (*Figure 2D*, bottom). Since we were interested in comparing EF occupancies between transcript classes, including unstable RNAs, we used Pol II PAR-CLIP normalization to calculate normalized EF PAR-CLIP occupancies, and used these for further analysis.

## EF localization along mRNA transcripts

To localize EFs on transcripts, we mapped the Pol II normalized PAR-CLIP occupancies onto transcripts in different classes (Materials and methods). We then calculated factor occupancies for 2532 mRNA transcripts that were filtered to reduce ambiguous signals from overlapping transcripts. We calculated heat maps with occupancies averaged around the transcript 5′-end, which corresponds to the transcription start site (TSS), and around the polyadenylation (pA) site (*Figure 3A*). The obtained profiles were also visible on individual transcripts (*Figure 3—figure supplement 1A*).

Generally, PAR-CLIP occupancies were high at the 5′-end of mRNAs and decreased shortly before the pA site, with few exceptions (*Figure 3A*). First, the histone methyltransferases Set2 and Dot1, for which the corresponding methylation marks accumulate in gene bodies (*Bannister et al., 2005*; *Pokholok et al., 2005*), showed more RNA-binding sites over transcript bodies. Second, Set1 crosslinked to mRNAs mainly near the beginning of transcripts, which was expected since Set1 and its methylation mark, H3K4me3, are observed in promoter-proximal regions of genes (*Ng et al., 2003*). Third, the kinases Ctk1 and Bur1 and their cyclin partners Ctk2 and Bur2 were enriched near the 5′-

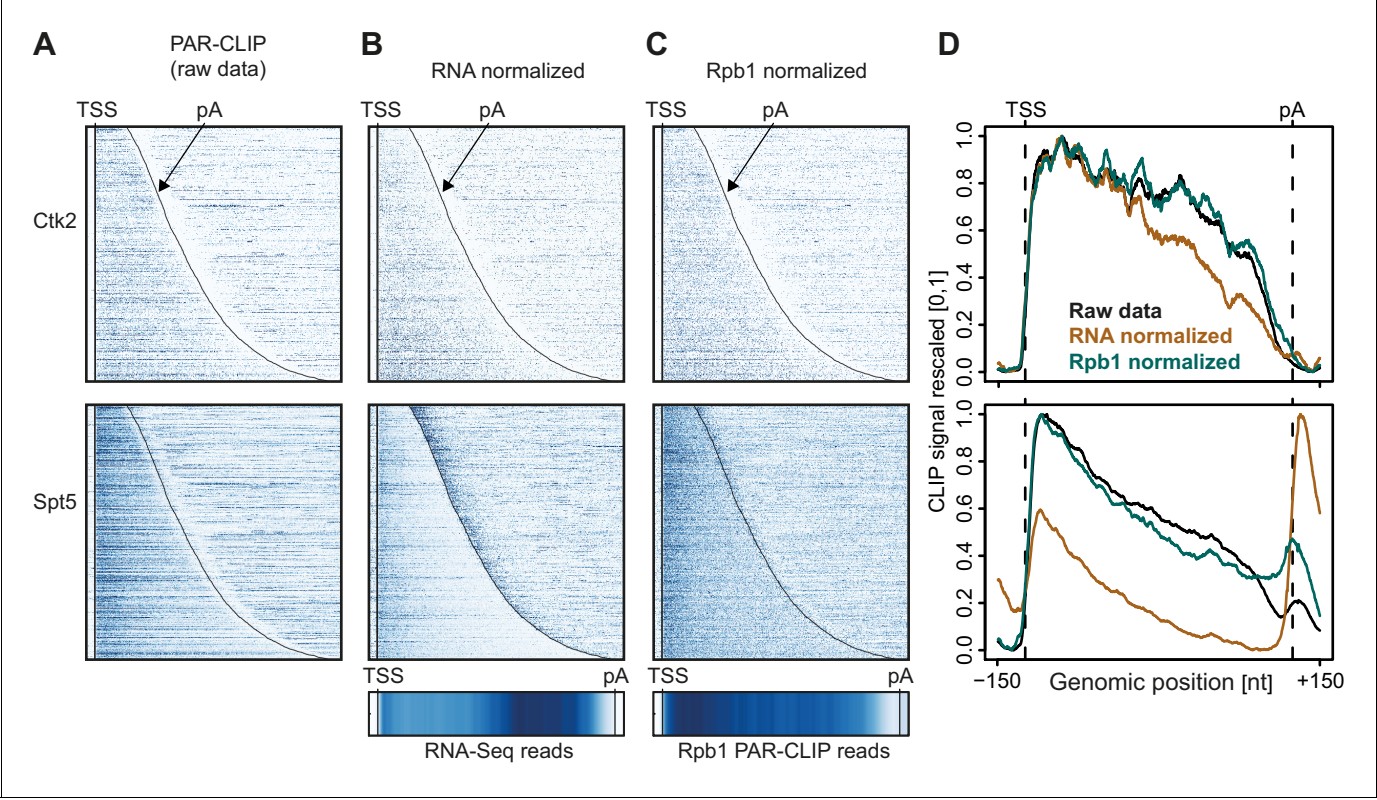

**Figure 2.** Normalization of PAR-CLIP data shown for two representative EFs, Ctk2 (top) and Spt5 (bottom). (**A**) Smoothed, raw RNA-binding strength as measured by the number of PAR-CLIP U-to-C transitions per U site for all mRNAs sorted by length and aligned at their RNA 5′-end (transcription start site, TSS). (**B**) Relative occupancy estimated by dividing the number of U-to-C transitions for each U site by the RNA-Seq signal at the corresponding genomic position for all mRNAs. A heat map showing the transcript-averaged RNA-Seq reads for all mRNAs scaled to the same length is shown below. (**C**) Relative occupancy estimated by dividing the number of U-to-C transitions for each U site by the Rpb1 PAR-CLIP signal at the corresponding genomic position for all mRNAs. A heat map showing the transcript-averaged Rpb1 PAR-CLIP reads for all mRNAs scaled to the same length is shown below. (**D**) Smoothed, raw and normalized PAR-CLIP signals as shown in A-C but averaged over all mRNAs. Before averaging RNA occupancy profiles were aligned at the RNA 5′-end and length-scaled such that the 5′-ends and pA sites coincided.

The following figure supplement is available for figure 2:

**Figure supplement 1.** Normalization of PAR-CLIP data shown for two representative EFs, Ctk2 (top) and Spt5 (bottom), at mRNAs (left) versus CUTs (right).

end but also in the transcript body. The 5′-peak for Bur1-Bur2 slightly preceded that of Ctk1-Ctk2. The three Paf1C subunits Cdc73, Ctr9 and Rtf1 showed similar occupancy profiles as the kinases but with a focused peak at the 5′-end. Fourth, Spt5 and Spt6 showed high PAR-CLIP occupancy at the 5′-end of mRNAs and decreased occupancy towards the pA site. This analysis revealed specific differences in EF localization on mRNAs, and additionally suggested that EFs bind nascent RNA during transcription.

## EFs bind nascent pre-mRNA

To test whether EFs interact with nascent pre-mRNA or with spliced, mature mRNA, we measured factor occupancies at introns, which are co-transcriptionally spliced out and subsequently degraded (*Carrillo Oesterreich et al., 2016*). All EFs cross-linked to introns (*Figure 3—figure supplement 1B*), indicating that they bind pre-mRNA. Most EFs bound to introns with a frequency that was comparable to that at exons, although Spt5 and Set1 showed slightly higher occupancy within introns, whereas Bur2, Set2 and Dot1 showed lower occupancy (*Figure 3—figure supplement 1B*). Taking into account that splicing generally occurs co-transcriptionally (*Kornblihtt et al., 2004*;

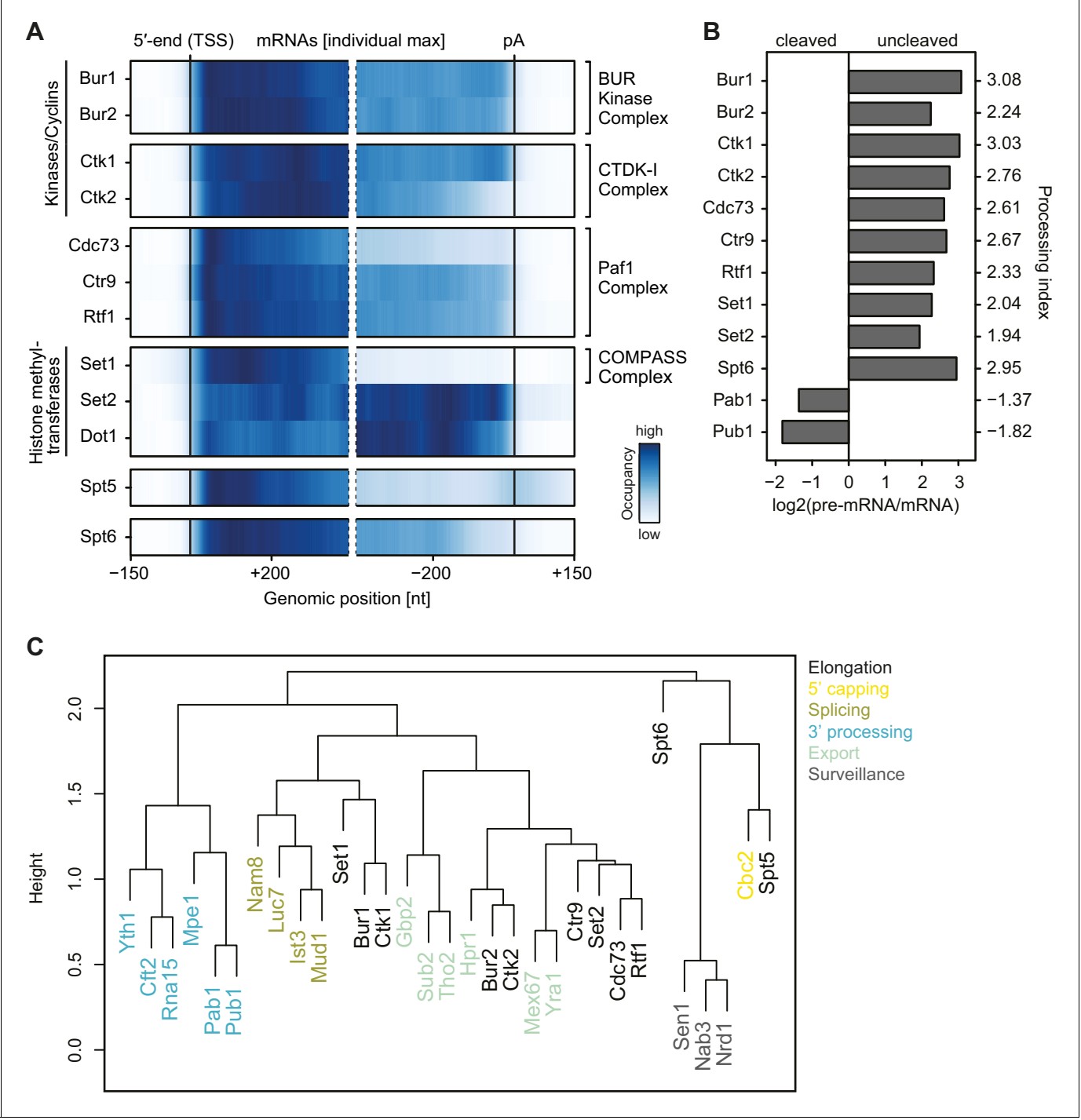

**Figure 3.** mRNA-binding profiles of EFs. (A) Smoothed, transcript-averaged Pol II normalized PAR-CLIP occupancy profiles of EFs centered around the transcript 5′-end (transcription start site, TSS) [−150 nt to +400 nt] and pA site [−400 nt to +150 nt] of a set of 2532 filtered mRNAs (compare *Figure 1*). Only factors with average RNA-binding occupancies > 2 fold above background are shown. The Spt5 PAR-CLIP profile reveals a peak downstream of the pA site that is discussed in detail elsewhere (*Baejen et al., 2017*). The color code shows the occupancy relative to the maximum occupancy per profile (dark blue). (B) EFs bind to pre-mRNA. Processing indices (PIs) measure preferential binding of factors to uncleaved pre-mRNA with respect to cleaved RNA, computed as log2 odds ratios uncleaved versus cleaved RNA bound by the factor (Materials and methods). The PIs for Pab1 and Pub1, as typical factors binding mature mRNA (*Baejen et al., 2014*), are shown for comparison. (C) Colocalization of factor crosslinking sites on transcripts. Euclidean distances between pairwise colocalization measures were subjected to average-linkage hierarchical clustering (Materials and methods). The cluster dendrogram shows similarities in crosslinking locations on transcripts between EFs and published RNA processing factors (*Baejen et al., 2014*; *Schulz et al., 2013*).

*Figure 3 continued on next page*

*Figure 3 continued*

The following figure supplement is available for figure 3:

**Figure supplement 1.** Non-averaged elongation factor RNA occupancies over mRNAs and introns.

*Tennyson et al., 1995*; *Listerman et al., 2006*), our data show that EFs interact with nascent pre-mRNA. However, only ~4% of yeast genes contain introns (*Qin et al., 2016*), preventing general statements related to all pre-mRNAs. We therefore calculated a processing index (PI) that measures preferential binding of factors to uncleaved pre-mRNA with respect to cleaved RNA (Materials and methods) (*Baejen et al., 2014*). All EFs showed positive PIs, indicating binding to pre-mRNA, in contrast to the negative PIs that we previously obtained for typical RNA binders of processed, mature mRNA, such as Pab1 and Pub1 (*Figure 3B*) (*Baejen et al., 2014*). We conclude that EFs preferentially interact with nascent pre-mRNA.

We next investigated where EFs localize on RNAs in relation to previously mapped mRNA biogenesis factors (*Baejen et al., 2014*). We determined the extent of factor colocalization by computing the average occupancy of factor A within ±20 nucleotides (nt) around RNA-binding sites of factor B and subjected the pairwise colocalization measures to hierarchical clustering (*Figure 3C*, Materials and methods). We found that Spt5 colocalizes with the Cbc2 subunit of the cap-binding complex, consistent with its recruitment during early elongation. Both Ctk1 and Bur1 colocalized with binding sites of Set1 and splicing factors. Paf1C subunits colocalized with Set2, whereas RNA 3'-processing and surveillance factors formed separate groups (*Figure 3C*). Together these data show a distinct distribution of EFs over RNAs, and suggested that EFs cooperate with other mRNA biogenesis factors during pre-mRNA binding.

## Most EFs preferentially interact with coding transcripts

We next analyzed our PAR-CLIP data for EF binding to non-coding Pol II transcripts including short-lived cryptic unstable transcripts (CUTs), which often arise from upstream antisense transcription of bidirectional promoters (*Wyers et al., 2005*; *Xu et al., 2009*). We selected CUTs with a minimum length of 350 nt and compared transcript-averaged RNA-binding occupancies between CUTs and mRNAs (see *Figure 4A*). This revealed that EFs bind to these transcript classes with distinct preferences. Spt5 was equally distributed between CUTs and mRNAs whereas Set1 preferentially bound mRNAs. All other EFs were depleted at CUTs relative to their mRNA occupancies (*Figure 4A*). This was essentially independent of RNA length (*Figure 4—figure supplement 1A*). Thus, most EFs preferentially crosslink to coding RNAs.

We then analyzed PAR-CLIP signals at bidirectional promoters, which produce mRNA in one direction and a CUT in the divergent direction (*Figure 4B*). We observed clear differences in PAR-CLIP signals for divergent directions. As in *Figure 4A*, Set1 and Spt5 showed high signals on CUTs and mRNAs (*Figure 4B*, top) whereas all other EFs bound exclusively to mRNAs (*Figure 4B*, bottom). These differences were also observed when the analysis was restricted to bidirectional promoters producing CUTs and mRNAs of similar lengths (*Figure 4—figure supplement 1B*).

How can some EFs distinguish between CUTs and mRNAs? We carried out motif analysis around the strongest PAR-CLIP sites for each EF using XXmotif (*Luehr et al., 2012*) and could not find any significantly enriched motifs, indicating that EFs bind RNA in a non-specific manner. We hypothesize that another RNA-binding factor blocks binding of EFs to CUTs. CUTs are rapidly degraded by a surveillance system, which includes Nrd1 (*Schulz et al., 2013*; *Vasiljeva et al., 2008*; *Steinmetz and Brow, 1996*). Nrd1 selectively binds to CUTs (*Figure 4C*) via motifs that are enriched in CUTs compared to mRNAs (*Schulz et al., 2013*). Binding of Nrd1 to CUTs might hinder RNA binding of some EFs, especially those which possess lower RNA binding affinity. This may explain how stable elongation complexes preferentially assemble on mRNAs.

## Chromatin association of EFs depends on RNA

We next investigated whether RNA binding of EFs contributes to their association with chromatin. Yeast cells were lysed and incubated with buffer containing RNases or with buffer only. Chromatin

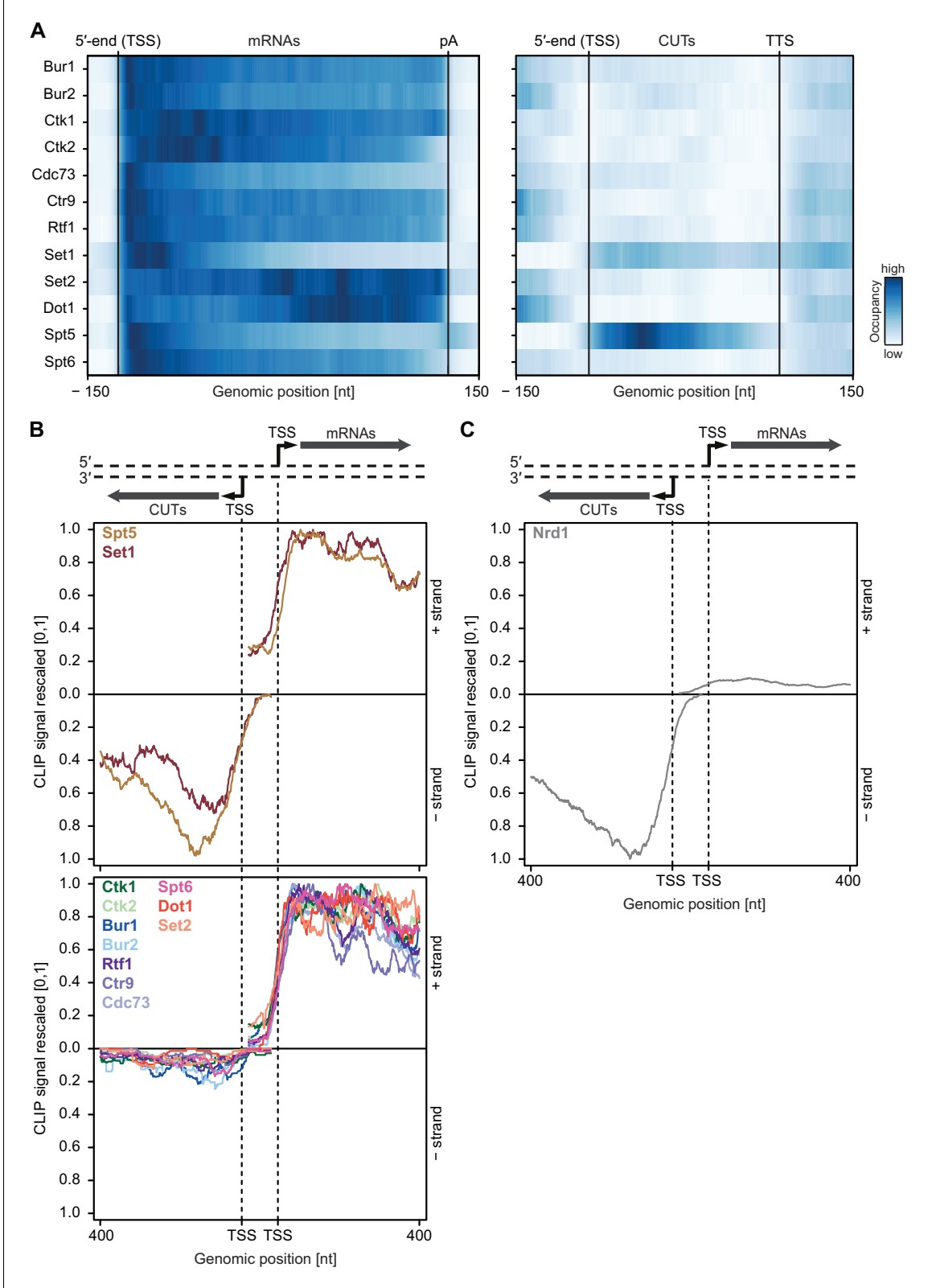

**Figure 4.** Asymmetric distribution of EFs at coding and non-coding transcripts. (**A**) PAR-CLIP occupancies over mRNAs (left) and non-coding CUTs (right). Smoothed, averaged Pol II normalized RNA occupancy profiles were aligned at the RNA 5'-end (transcription start site, TSS) and scaled to a common length. The color code shows the occupancy relative to the maximum occupancy per factor over both transcript classes (dark blue) (see also *Figure 4—figure supplement 1A*). (**B**) and (**C**) PAR-CLIP occupancies at selected bidirectional promoters. Smoothed, averaged Pol II normalized RNA

*Figure 4 continued on next page*

*Figure 4 continued*

occupancy profiles for sense mRNA (right) and divergent CUT (left) were centered around their 5'-end (TSS) [−75 nt to +400 nt]. We considered only bidirectional promoters producing mRNAs and CUTs that did not overlap with any other transcripts in the depicted region. After normalization, average mRNA and CUT profiles were rescaled, setting the maximum occupancy to one and the minimum occupancy to 0 (see also ***Figure 4—figure supplement 1B and C***).

The following figure supplement is available for figure 4:

**Figure supplement 1.** Asymmetric distribution of EFs at coding and non-coding transcripts of similar length.

was isolated and associated protein factors were detected by Western blotting (Materials and methods). We found that RNase treatment strongly decreased the levels of chromatin-associated enzymes Set1, Set2, Dot1, Bur1, Ctk1, and the cyclins Bur2 and Ctk2 (***Figure 5***). Thus, RNA stabilizes chromatin association of these factors. Two non-enzymatic EFs also depended on RNA for chromatin association, although less strongly. With respect to Paf1C, Rtf1 was partially lost upon RNase treatment, whereas Leo1 and Paf1 were not significantly affected (***Figure 5***). Spt5 binding to chromatin also depended on RNA, whereas Spt6 was not significantly affected by RNase treatment (***Figure 5***). These data are generally consistent with our PAR-CLIP results. The discrepancies between RNA-dependent chromatin association and PAR-CLIP results for Spt6 (high PAR-CLIP signal versus RNA-independent chromatin binding) and Dot1 (low PAR-CLIP signal versus strong RNA-dependent chromatin binding) can be explained by additional protein-protein interactions, and by the dependence of PAR-CLIP on the concentration of the RNA-interacting protein in the cell (***Chong et al., 2015***; ***Kulak et al., 2014***).

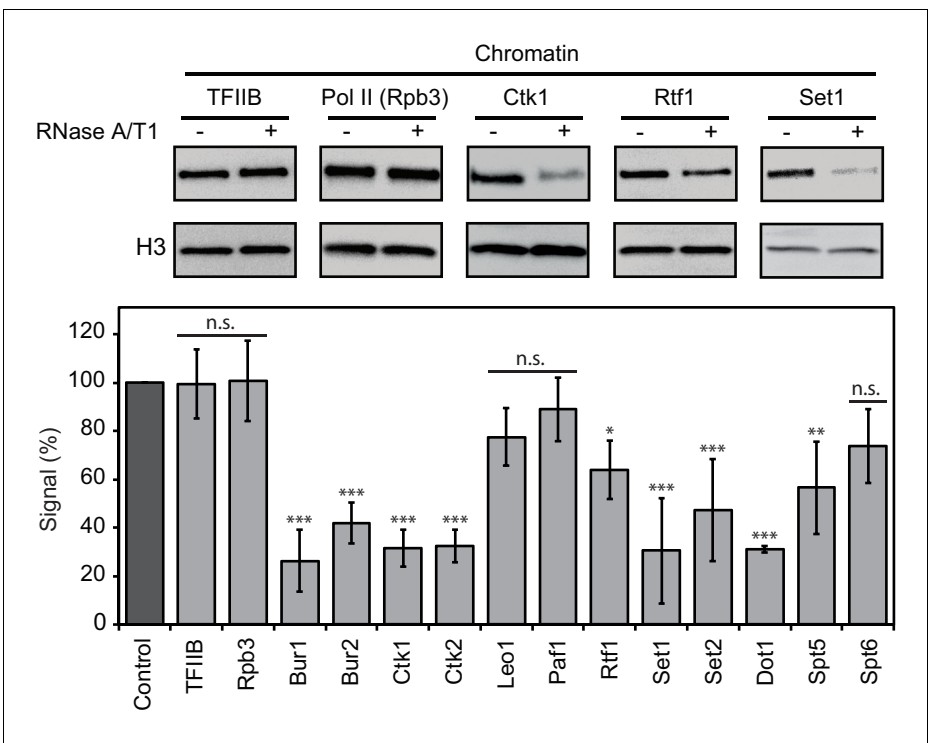

**Figure 5.** Chromatin association of EFs depends on RNA. Western blot analysis (top) and quantitative densitometry (bottom) of exemplary EFs bound to chromatin before and after treatment with RNase A/T1 mix. H3 was used as loading control. Densitometry data are expressed as mean ± SD from two to three independent experiments. *p<0.05; **p<0.01; ***p<0.001; n.s. = not significant (one-way ANOVA Dunnett post-hoc test).

As a negative control, we subjected TFIIB to the RNase assay. We observed no differences in chromatin binding after RNase treatment (*Figure 5*), consistent with recruitment of TFIIB to DNA during transcription initiation (*Sainsbury et al., 2015*). Also as expected, RNase treatment did not affect association of Pol II with chromatin, showing that the observed losses of EFs from chromatin upon RNase treatment were not due to a loss of Pol II (*Figure 5*). These controls and the above results show that the association of many EFs with chromatin depends on RNA.

## Ctk1 kinase complex binds RNA in vitro

The observed RNA-EF crosslinking in vivo and the RNA-dependent chromatin association data strongly suggested that EFs can directly bind RNA. To investigate this in vitro, we prepared one EF complex in recombinant form. We chose the prominent Ser2 kinase complex CTDK-I that comprises Ctk1, Ctk2, and the small subunit Ctk3 (*Mühlbacher et al., 2015*; *Sterner et al., 1995*). CTDK-I is the main yeast kinase responsible for phosphorylating the Pol II CTD at Ser2 (*Patturajan et al., 1999*; *Cho et al., 2001*), and this is a decisive event in establishing a mature Pol II elongation complex. Further, RNA-dependent chromatin association of Ctk1 and Ctk2 were most unexpected, as for several other EFs RNA interactions were already reported (compare introduction).

We co-expressed recombinant Ctk1, Ctk2, and Ctk3 in insect cells and purified a complete, intact CTDK-I complex (Materials and methods, *Figure 6A*). We then tested the purified CTDK-I complex for its kinase activity using a purified GST-CTD construct and dephosphorylated full-length *S. cerevisiae* Pol II (Materials and methods). Both the GST-CTD and the Rpb1 subunit of Pol II were readily phosphorylated by CTDK-I at the Ser2 position in vitro (*Figure 6B,C*), showing that our purified CTDK-I complex was active.

We then tested the purified CTDK-I complex for RNA binding in vitro. We performed fluorescence anisotropy titration experiments using single-stranded (ss) RNA oligonucleotides with 45% or 24% GC content and bearing a 5′ FAM label (*Figure 6D,E*). CTDK-I bound both ssRNAs with similar affinities (*Figure 6D*). We also tested U- or A-rich sequences for association with CTDK-I and found some preference for U-rich RNA (*Figure 6E*, *Figure 6—figure supplement 1*). Fitting the data with binding curves by linear regression resulted in apparent $K_d$'s in the nanomolar range (*Figure 6D,E*). All experiments were done in the presence of tRNA as competitor, indicating that flexible, single-stranded nucleic acids are preferentially bound. Consistent with this, CTDK-I bound to duplex DNA much more weakly (dsDNA, *Figure 6E*). These experiments show that the EF complex CTDK-I binds to single-stranded RNA in vitro, consistent with direct EF-RNA interactions in vivo.

## Evidence that RNA contributes to EF recruitment

We also measured gene occupancies of EFs using ChIP-Seq and compared them with our PAR-CLIP occupancies (*Figure 7*). The obtained ChIP-Seq data sets were highly reproducible (*Figure 7—figure supplement 1*). For comparability with PAR-CLIP data, we collected ChIP-Seq data, although ChIP data are available for single genes or genome-wide using various other techniques or set-ups (*Keogh et al., 2003*; *Kim et al., 2004*; *Kizer et al., 2005*; *Krogan et al., 2003b*; *Liu et al., 2005*; *Mayer et al., 2010*; *Ng et al., 2003*; *Pokholok et al., 2005*; *Weiner et al., 2015*). Metagene analysis of our ChIP-Seq data revealed that EF occupancy increased within 100–600 bp downstream of the TSS, and was generally high in gene bodies (*Figure 7*, red lines). In contrast, PAR-CLIP results showed that EFs interacted with RNA already from around 20 nt downstream of the capped 5′-end of mRNAs (*Figure 7*, blue lines). This difference was most pronounced for Set2, which occupies transcripts at the 5′-end but showed peak levels of genome association only in the downstream region, with peak levels 450–300 bp upstream of the pA site. These results are consistent with the idea that RNA contributes to EF recruitment to transcribed genes, and that the contribution of RNA-based recruitment differs for different EFs.

Comparison of our histone methyltransferase PAR-CLIP data sets with ChIP-Seq data of the corresponding methylation marks (*Figure 7*, left, orange lines) provides further support of the model that RNA binding can contribute to EF recruitment to transcribed regions. In the direction of transcription, the PAR-CLIP signals for methyltransferases increased first, followed by an onset of ChIP-Seq signals for the respective histone methylation marks, which in turn preceded the increase in ChIP-Seq signals for the enzymes (*Figure 7*, left). This sequence of signal onsets is consistent with the

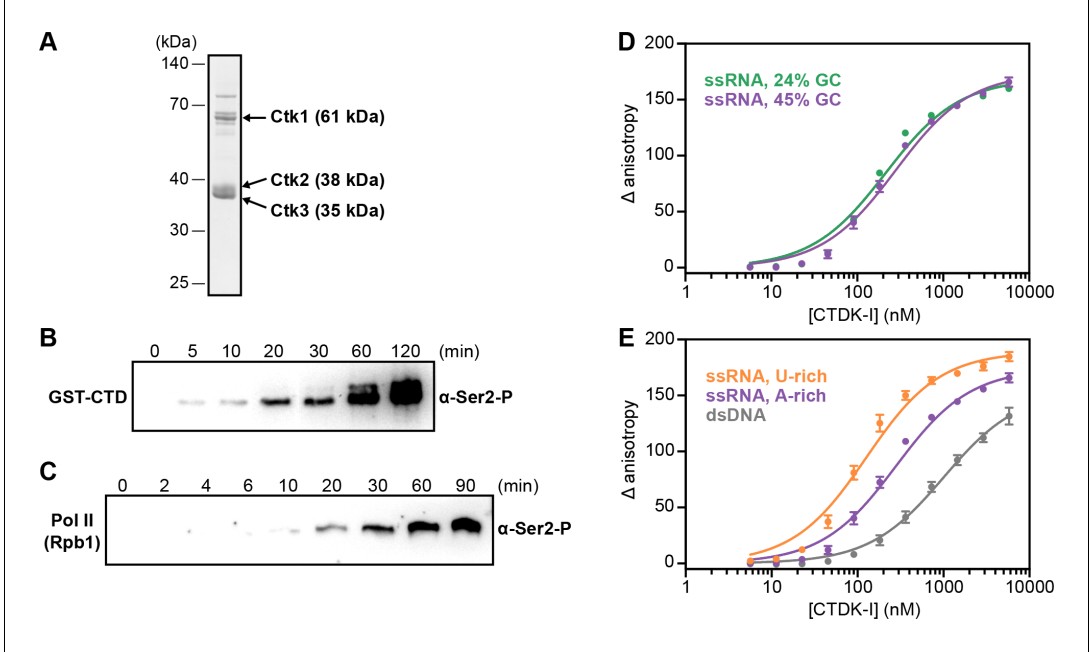

**Figure 6.** Recombinant CTDK-I complex is active and binds RNA in vitro. (**A**) The three-subunit CTDK-I complex from *S. cerevisiae* was recombinantly expressed in insect cells and purified to homogeneity. The purified complex was run on a 4–12% gradient sodium dodecyl sulfate polyacrylamide gel electrophoresis (SDS-PAGE) and stained with Coomassie blue. (**B**) Purified human GST-CTD (10 µM) was incubated with 0.4 µM CTDK-I and 3 mM ATP. Time points were taken at 0 (no ATP), 5, 10, 20, 30, 60 and 120 min and CTDK-I activity was determined by western blot analysis using an antibody that recognizes the Ser2 phosphorylated form of the CTD of Pol II. Molecular mass of GST-CTD is ~70 kDa. (**C**) Purified and dephosporylated Pol II (2 µM) from *S. cerevisiae* was incubated with 0.4 µM CTDK-I and 3 mM ATP. Time points were taken at 0 (no ATP), 2, 4, 6, 10, 20, 30, 60 and 90 min and CTDK-I activity was determined as in (**B**). Molecular mass of the CTD containing subunit of Pol II, Rpb1, is ~200 kDa. (**D**) Increasing concentrations (0–5.8 µM) of the complete CTDK-I kinase complex were incubated with 8 nM of a 24% GC (green line; $K_{d,app}$(nM) = 210 ± 18) and with a 45% GC (purple line; $K_{d,app}$(nM) = 277 ± 21) ssRNA sequences. Binding was determined by relative change in fluorescence anisotropy. Data was fit with a single site binding equation. Error bars reflect the standard deviation from three experimental replicates. (**E**) Increasing concentrations (0–5.8 µM) of the complete CTDK-I kinase complex were incubated with 8 nM of a U-rich ssRNA (orange line; $K_{d,app}$(nM) = 123 ± 10), an A-rich ssRNA (purple line; $K_{d,app}$(nM) = 277 ± 21) and a dsDNA (grey line; $K_{d,app}$(nM) = 1007 ± 67) sequences. Binding strength, data fitting and standard deviation was determined as in (**D**).

The following figure supplement is available for figure 6:

**Figure supplement 1.** Recombinant and active CTDK-I complex binds preferentially U-rich ssRNA in vitro.

model that these EFs are recruited by nascent RNA and then modify histones as Pol II moves downstream.

To test the model of RNA-based recruitment for a particular factor, we investigated whether Set1 gene occupancy depends on the N-terminal region of the protein that contains two RNA recognition motifs (RRMs, residues 247–375 and 376–579) that bind RNA in vitro (*Trésaugues et al., 2006*). We performed ChIP-qPCR analysis for Set1 in a strain lacking its N-terminal residues 1–579 (ΔRRM-Set1-TAP, Materials and methods) (*Figure 8*). Additionally, we carried out Set1 ChIP-qPCR in a mutant lacking Paf1 (ΔPaf1 Set1-TAP) because the Paf1 complex was shown to contribute to Set1 recruitment (*Krogan et al., 2003a*). We compared Set1 gene occupancy levels of both mutant strains (ΔRRM-Set1-TAP and ΔPaf1 Set1-TAP) with the full-length protein occupancy in a Set1-TAP strain. All strains expressed similar levels of Set1 and the Pol II subunit Rpb3 (*Figure 8A*). We analyzed protein occupancy at different genomic regions of four housekeeping genes (*Figure 8B*) and at a non-transcribed region of chromosome V. Gene regions within the first 1000 bp downstream of the TSS showed a severe decrease in ΔRRM-Set1 occupancy (*Figure 8C*; genomic regions 1, 2, 4 and 6). Similarly, we also detected a decrease in Set1 occupancy in the absence of Paf1, confirming the role of Paf1C in Set1 recruitment (*Figure 8C*). These results indicate that Set1 recruitment not only depends on the Paf1 complex, but also on binding to nascent RNA. Taken together, several lines of evidence

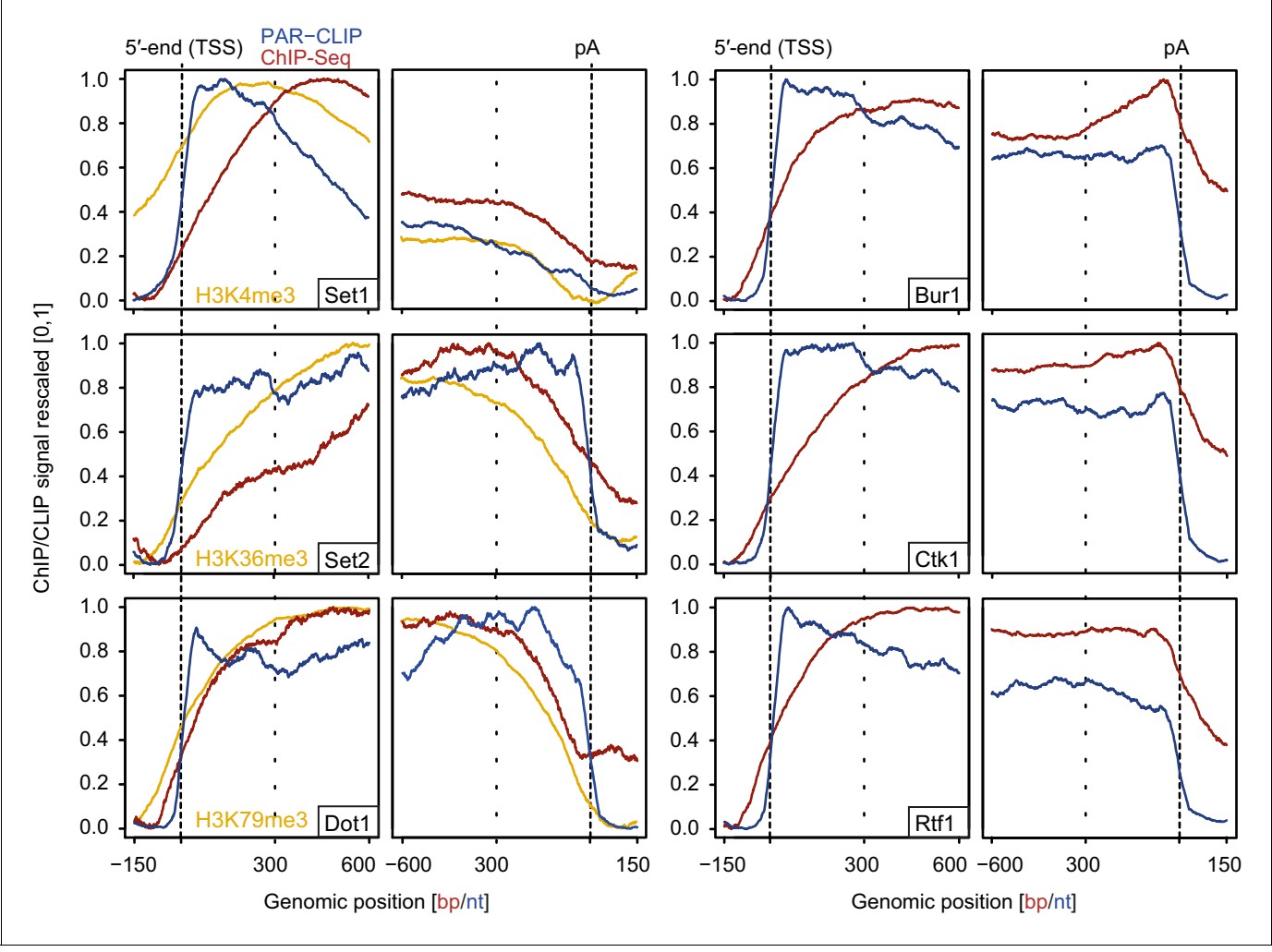

**Figure 7.** Comparison of PAR-CLIP and ChIP-Seq occupancy profiles. Averaged ChIP-Seq (red) and PAR-CLIP (blue) occupancy profiles of EFs and ChIP-Seq of the histone marks H3K4me3, H3K79me3 and H3K36me3 (yellow) centered around TSSs [−150 bp to +600 bp] and pA sites [−600 bp to +150 bp] individually normalized to range between 0 and 1.

The following figure supplement is available for figure 7:

**Figure supplement 1.** Comparison of replicate measurements for ChIP-Seq.

presented here strongly suggest that interactions of EFs with nascent RNA contribute to EF recruitment to actively transcribed genes in vivo.

## Discussion

Here we present a large set of system-wide occupancy data for yeast transcription elongation factors on RNA (PAR-CLIP) and DNA (ChIP-Seq), and complementary biochemical data. The remarkable finding from our work is that many EFs interact with nascent RNA in vivo. Additional in vitro results support these findings and indicate that RNA can contribute to EF recruitment and the stability of the transcription elongation complex. For Set1 we further demonstrate that the two RNA recognition motifs are required for Set1 recruitment to genes in vivo. These results extend our understanding of how the transcription elongation complex is assembled and maintained on active genes. The emerging view from our data is that nascent RNA contributes to EF recruitment and elongation complex stability to different extents for different EFs. We note that our results do not reveal whether all

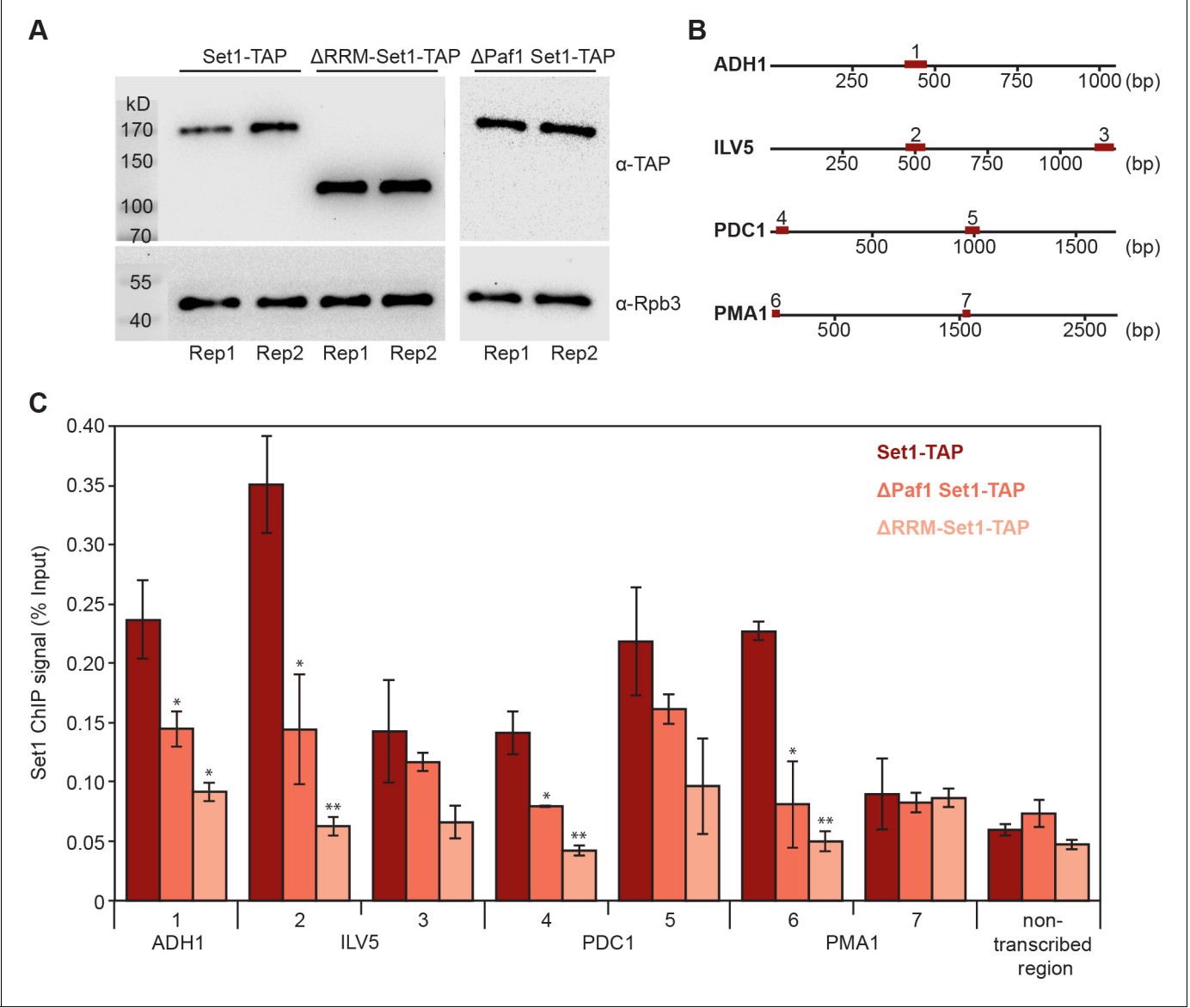

**Figure 8.** Deletion of Set1 RRMs impairs its recruitment to genes. (**A**) Western blot analysis of Set1-TAP (top) and Rpb3 (bottom) in a Set1-TAP strain (left), a strain lacking the first 579 amino acids of Set1 (ΔRRM-Set1-TAP; middle) and a ΔPaf1 Set1-TAP strain (right); bands are shown for biological duplicates of yeast cell cultures before formaldehyde crosslinking. Set1 was detected using an antibody directed against its C-terminal TAP tag. As a control, Pol II was detected using an antibody against the Rpb3 subunit in all three strains. (**B**) Schematic localization of gene regions analyzed via ChIP-qPCR. Set1 recruitment was monitored at one gene region of *ADH1* (1) and two different gene regions of *ILV5* (2 and 3), *PDC1* (4 and 5) and *PMA1* (6 and 7). (**C**) ChIP analysis reveals that Set1 occupancy is reduced in *ΔPaf1* cells (*ΔPaf1* Set1-TAP) as well as in a truncated version of Set1 that lacks its RRM domains (*ΔRRM-Set1-TAP*). ChIP data are expressed as mean ± SD from two independent experiments. *p<0.05; **p<0.01 (two sample t-test).

EFs studied here are initially recruited by RNA, and which EFs establish RNA interactions only after they have been recruited by alternative interactions, although EF binding in the very 5′-region of transcripts argues for a RNA-based recruitment model.

Our results also elucidate the long-standing question how the yeast CTD Ser2 kinases Ctk1 and Bur1, which are essential for transcription elongation, are recruited to transcribing Pol II. The Pol II Ser2 kinases give rise to strong PAR-CLIP signals and their chromatin association is strongly dependent on RNA. In addition, we show that purified CTDK-I complex strongly binds to RNA in vitro. This all indicates that nascent RNA plays an important role in recruiting Ser2 kinases to transcribing Pol II. Binding of the Ser2 kinases near the RNA 5′-end is consistent with stabilization of these kinases on

the elongation complex by the cap-binding complex (*Hossain et al., 2013*; *Lidschreiber et al., 2013*). A model of kinase recruitment by capped RNA predicts that these enzymes are lost from the transcribing enzyme upon RNA cleavage at the pA site, and this is indeed observed by ChIP-Seq. In conclusion, RNA-based recruitment of Ser2 kinases explains why Ser2 phosphorylation of the CTD is restricted to transcribing polymerases, whereas free or initiating polymerases are not phosphorylated at Ser2 residues.

How can some EFs bind both RNA and Pol II? EFs are generally modular and contain multiple domains that can be involved in RNA or protein interactions. However, the same domain can mediate both RNA and protein interactions, as documented for the RNA export factor Yra1, which contains a RNA recognition motif (RRM) domain that binds both RNA and the phosphorylated CTD (*MacKellar and Greenleaf, 2011*). Set1 contains two adjacent RRM domains (*Trésaugues et al., 2006*), and Set2 contains a SRI domain that binds the phosphorylated CTD (*Dengl et al., 2009*; *Sun et al., 2010*; *Yoh et al., 2007*; *MacKellar and Greenleaf, 2011*), but may also bind RNA. The three Paf1C subunits that bind RNA in vivo, namely Cdc73, Ctr9 and Rtf1, also interact with the phosphorylated CTD and the phosphorylated C-terminal region (CTR) of Spt5 in vitro (*Qiu et al., 2012*). Rtf1 contains a positively charged Plus-3 domain (*Finn et al., 2014*) that binds the phosphorylated CTR (*Wier et al., 2013*) and single-stranded nucleic acids (*de Jong et al., 2008*). We predict that many EFs contain domains that can interact with RNA or with the phosphorylated CTD or CTR, which resemble RNA in its flexible nature and negative charge. Whereas for some EFs binding to RNA or the CTD may be mutually exclusive, others can bind both Pol II and RNA at the same time, for example Spt5. Due to a lack of solubility of individually expressed EF subunits, and the difficulty of preparing EF complexes in recombinant and pure form in large quantities, we had to limit our in vitro RNA-binding analysis to CTDK-I.

Finally, we predict that RNA-based recruitment of EFs provides a missing link in our understanding of how the transcription cycle is coordinated. When the initiation complex assembles at the promoter, TFIIH phosphorylates Ser5 residues in the CTD and this enables recruitment of the capping enzyme (*Cho et al., 1997*; *Fabrega et al., 2003*; *Rodriguez et al., 2000*; *Schroeder et al., 2000*; *Schwer and Shuman, 2011*). The nascent RNA then receives a 5′-cap (*Martinez-Rucobo et al., 2015*), and capped RNA could then help to recruit EFs. The requirement for a cap on RNA befits the observation that Ser5 phosphorylation is needed for high gene occupancy with some EFs (*Qiu et al., 2012*, *2009*, *2006*; *Ng et al., 2003*). RNA-based recruitment of the major Ser2 kinase, Ctk1, would then lead to CTD phosphorylation on Ser2 residues and stable binding of other EFs. Eventually, transcription of a pA site triggers RNA cleavage, and this would facilitate loss of many RNA-bound EFs and render the polymerase prone to transcription termination. Thus, the transcribing Pol II complex may be viewed as a self-organizing system that is encoded in the DNA, but only realized on the level of RNA, which plays crucial roles in complex assembly and disassembly.

## Materials and methods

### Strains and antibodies

*Saccharomyces cerevisiae* (*Sc*) BY4741 strains containing C-terminally TAP-tagged genes (Open Biosystems, Germany) were tested for expression of the correctly tagged protein using the Peroxidase Anti-Peroxidase (PAP; Sigma, P1291, St. Louis, MO) antibody. To obtain the ΔPaf1 Set1-TAP strain, a ΔPaf1-KanMX6 cassette, amplified from the pFA6a-KanMX6 vector (*Supplementary file 1*), was introduced by homologous recombination into a Set1-TAP strain. A DNA fragment coding for Set1 residues 580 to 1080 and a C-terminal TAP tag was amplified from genomic DNA from a Set1-TAP strain (*Supplementary file 1*) and transformed into a ΔSet1 strain (Open Biosystems) by homologous recombination. Additional antibodies used were anti-Histone H3 (HRP; Abcam, ab21054, UK), anti-Histone H3 (tri methyl K4; Abcam, ab8580), anti-Histone H3 (tri methyl K36; Abcam, ab9050), anti-Histone H3 (tri methyl K79; Abcam, ab2621), IgG from rabbit serum (directed against the protein A content of the C-terminal TAP tag of proteins; Sigma, I5006), anti-rat IgG (HRP; Sigma, A9037) and anti-Ser2P (3E10; kindly provided by Dirk Eick [*Chapman et al., 2007*]).

## PAR-CLIP of *S. cerevisiae* proteins

PAR-CLIP was performed as described (*Baejen et al., 2014*; *Schulz et al., 2013*), with modifications. The full protocol is described here for convenience. Yeast cells expressing the TAP-tagged protein were grown at 30°C to $OD_{600}$~0.5 in minimal medium (CSM mixture, Formedium, UK) containing 10 mg/L uracil, 100 µM 4-thiouracil (4tU) and 2% glucose. 4-Thiouracil was added to a final concentration of 1 mM and cells were grown further for 4 hr. Following RNA labeling, cells were harvested, resuspended in 1× PBS and UV-irradiated on ice with an energy dose of 12 J/cm$^2$ at 365 nm under continuous shaking. Cells were harvested, resuspended in lysis buffer (50 mM Tris-HCl pH 7.5, 100 mM NaCl, 0.5% sodium deoxycholate, 0.1% SDS, 0.5% NP-40), and disrupted by bead beating (FastPrep−24 Instrument, MP Biomedicals, LLC., France) in the presence of 1 mL of silica-zirconium beads (Roth, Germany) for 40 s at 4 m/s, followed by an incubation of the sample for 1 min on ice. This was repeated eight times. The success of the cell lysis was monitored by photometric measurements and the cell lysis efficiency was usually >80%. Samples were solubilized for 1 min via sonication with a Covaris S220 instrument (Covaris, UK) using following parameters: Peak Incident Power (W): 140; Duty Factor: 5%; Cycles per Burst: 200. The lysate was cleared by centrifugation. Immunoprecipitation was performed on a rotating wheel overnight at 4°C with rabbit IgG-conjugated Protein G magnetic beads (Invitrogen, Germany). Beads were washed twice in wash buffer (50 mM Tris-HCl pH 7.5, 500 mM NaCl, 0.5% sodium deoxycholate, 0.1% SDS, 0.5% NP-40) and once in T1 buffer (50 mM Tris-HCl pH 7.5, 2 mM EDTA). Immunoprecipitated and crosslinked RNA was partially digested with 50 U of RNase T1 per mL for 20 min at 25°C and 400 rpm. Beads were washed twice in T1 buffer and phosphatase reaction buffer (50 mM Tris-HCl pH 7.0, 1 mM MgCl$_2$, 0.1 mM ZnCl$_2$). For dephosphorylation, 1× antarctic phosphatase reaction buffer (NEB, Germany) with 1 U/µL of antarctic phosphatase and 1 U/µL of RNase OUT (Invitrogen) were added and the suspension was incubated at 37°C for 30 min and 800 rpm. Beads were washed once in phosphatase wash buffer (50 mM Tris-HCl pH 7.5, 20 mM EGTA, 0.5% NP-40) and twice in polynucleotide kinase (PNK) buffer (50 mM Tris-HCl pH 7.5, 50 mM NaCl, 10 mM MgCl$_2$). Beads were resuspended in 1 × T4 PNK reaction buffer A (Fermentas, Germany) with a final concentration of 1 U/µL T4 PNK and 1 U/µL RNase OUT. Phosphorylation of PAR-CLIP samples was performed using either 1 mM ATP per mL (cold-labeling) or 0.5 µCi of gamma-32-P-ATP per mL (radioactive labeling). The bead suspension was incubated for 1 hr at 37°C and 800 rpm and washed in PNK buffer. For visualization of protein-RNA interactions, the radioactively labeled samples were subjected to SDS-PAGE analysis. Radioactive RNA-protein bands were detected with the Typhoon FLA 9500 instrument (Typhoon, Sweden).

## PAR-CLIP library preparation and high-throughput sequencing

For 3′ adaptor ligation, beads were resuspended in 1 × T4 RNA ligase buffer (NEB) containing 10 U/µL T4 RNA ligase 2 (KQ) (NEB, M0373), 10 µM 3′ adaptor (5′ 5rApp-TGGAATTCTCGGG TGCCAAGG-3ddC 3′ (IDT, Inc., Coralville, IA)), 1 U/µL RNase OUT, and 15% (w/v) PEG 8000. The bead suspension was incubated for 18 hr at 16°C and 600 rpm. Beads were washed in PNK buffer to remove unligated adapters. For 5′ adaptor ligation, beads were resuspended in 1 × T4 RNA ligase buffer (NEB) containing 6 U/µL T4 RNA ligase 1 (NEB), 10 µM 5′ adaptor (5′ 5InvddT-GUUCAGAG UUCUACAGUCCGACGAUCNNNNN 3′, IDT), 1 mM ATP, 1 U/µL RNase OUT, 5% (v/v) DMSO, and 10% (w/v) PEG 8000. The suspension was incubated for 4 hr at 24°C and 600 rpm. Beads were washed twice in PNK buffer, and twice in proteinase K buffer (50 mM Tris-HCl pH 7.5, 6.25 mM EDTA, 75 mM NaCl). Beads were boiled twice at 95°C for 5 min in proteinase K buffer with 1% SDS and eluted RNA-protein complexes were treated with 1.5 mg/mL proteinase K (NEB) for 2 hr at 55°C. RNA was recovered by acidic phenol/chloroform extraction followed by ethanol precipitation supported by addition of 0.5 µL GlycoBlue (Invitrogen) and 100 µM RT primer (5′ CCTTGGCACCC-GAGAATTCCA 3′). Reverse transcription was performed for 1 hr at 44°C using SuperScript III RTase. For PCR amplification, NEXTflex barcode primer and universal primer and Phusion HF master mix (NEB) were added. After PCR amplification, cDNA was purified and size-selected on a precast 4% E-Gel EX Agarose Gel (Invitrogen), quantified on an Agilent 2200 TapeStation instrument (Agilent Technologies, Germany), and sequenced on an Illumina HiSeq 1500 sequencer (Illumina, Inc., San Diego, CA). We performed three independent biological replicates for Bur1, Cdc73, Leo1 and Rtf1 and two independent biological replicates for TFIIB, Spt6, Set2, Dot1, Set1, Paf1, Ctr9, Ctk2, Ctk1 and Bur2.

## PAR-CLIP data processing

Data quality control and mapping was performed as described (*Baejen et al., 2014*) with some modifications. Briefly, quality-trimmed reads are aligned to the *S. cerevisiae* genome (sacCer3, version 64.2.1) using the short read aligner STAR (version 2.5.2b; options: –outFilterMultimapNmax 1, –outFilterMismatchNmax 1, –scoreDelOpen −10000, –scoreInsOpen −10000, –alignSJoverhangMin 10000, –alignSJstitchMismatchNmax 0 0 0 0 [*Dobin et al., 2013*]). The resulting SAM files are then converted into BAM and PileUp files using SAMTools (*Li et al., 2009*).

We calculated the P-values for true crosslinking sites as described (*Baejen et al., 2014*). Briefly, we had to quantitatively model the null hypothesis, that is, the probability that the T-to-C mismatches observed in reads covering a certain T nucleotide in the genome were not caused by crosslinks between the immunoprecipitated factor and RNA but are due to the other sources of mismatches. Owing to the exquisite sensitivity of our experimental PAR-CLIP procedure, we could set a very stringent P-value cut-off of 0.005 and a minimum coverage threshold of two. For true crosslinking sites passing our stringent thresholds, the PAR-CLIP-induced T-to-C transitions strongly dominate over the contributions by sequencing errors and SNPs. For any given T site in the transcriptome, the number of reads showing the T-to-C transition is proportional to the occupancy of the factor on the RNA times the concentration of RNAs covering the T site. Therefore, the occupancy of the factor on the RNA is proportional to the number of reads showing the T-to-C transition divided by the concentration of RNAs covering the T site. This concentration was estimated either from the RNA-Seq read coverage measured under comparable conditions as described (*Baejen et al., 2014*) or by the read coverage obtained from a Rpb1 PAR-CLIP experiment (this study) and was used to obtain normalized occupancies. We compared RNA and Pol II (Rpb1) normalized occupancy profiles and found that the latter were less prone to biases introduced due to difficulties in measuring unstable RNA species, including CUTs, introns and nascent transcripts downstream of the pA site.

## PAR-CLIP data analysis

For transcript annotation, we used the recent TIF-Seq data from (*Pelechano et al., 2013*) to derive TSS and pA site annotations for 5578 coding genes. TSS and TTS positions of non-coding RNAs were taken from (*Xu et al., 2009*) for CUTs and from the Saccharomyces Genome Database (SGD, version = R64–2–1) for snoRNAs. Annotated transcripts were distance-filtered for downstream analysis to reduce ambiguous signals from overlapping transcripts. Unless stated otherwise, mRNA transcripts were selected to be at least 150 nt away from neighboring transcripts on the same strand. Unless stated otherwise, mRNAs and CUTs were selected to be 800–5000 nt and 350–1500 nt long, respectively. Bidirectional promoters were selected as follows: distance between TSS of mRNAs and divergent CUTs was smaller than 350 bp. Moreover, only mRNAs and CUTs that did not overlap with any other transcripts in the region from their TSS to 400 nt downstream on the same strand were considered.

To generate transcript class-averaged heat maps and profiles, transcripts were aligned at their 5'-end ('TSS') and pA sites and either scaled to the same length (median) or cut around the TSS and pA sites before taking the average RNA-binding occupancy at each genomic position. Average occupancies were smoothed (sliding window averaging using a 61 nt window, 30 nt to either side of the current position) and for each factor individually re-scaled between 0 (minimum signal) and 1 (maximum signal) for all figures but *Figure 1—figure supplement 1A*, for which all factors were globally scaled to show the relative strength of factor binding. To compare averaged RNA-binding occupancies between transcript classes, they were scaled together by setting min (transcript class 1, transcript class 2) to 0 and max (transcript class 1, transcript class 2) to 1 (*Figure 2—figure supplement 1*, *Figure 4* and *Figure 4—figure supplement 1*).

For generation of non-averaged heat maps of filtered mRNAs (*Figure 2* and *Figure 3—figure supplement 1A*) transcripts were sorted by length and aligned at their 5'-end ('TSS'). Smoothed occupancies were binned in cells of 20 nucleotide positions times 10 transcripts to avoid aliasing effects due to limited resolution of the plots. The color code displays the occupancy of the PAR-CLIPped factor (with the 97% quantile of these bins scaled to 1). In *Figure 3—figure supplement 1B*, all introns (SGD annotation) with lengths between 150 and 650 nt were aligned at the 5'-splice site (5'SS) and the occupancy of each intron is displayed without binning in either x or y direction.

PAR-CLIP processing indices (PIs) (**Figure 3B**) were calculated essentially as described (**Baejen et al., 2014**; **Schulz et al., 2013**). We assume that read counts (not crosslinking sites) $N^{down}$ downstream of a pA site can only occur from pre-mRNAs, $N^{down} = N^{prem}$, whereas read counts $N^{up}$ upstream of a pA site are a mixture of mature mRNA counts $N^{mat}$ and pre-mRNA counts $N^{prem}$. Therefore, $N^{up} = N^{mat} + N^{prem}$. For increased robustness with regard to different transcript isoforms and uncertainties in the exact location of pA sites, we computed $N_i^{up}$ and $N_i^{down}$ as average of the read counts for each transcript $i$ of a given annotation $A$:

$$N_i^{up} = 1/50 \sum_{j=pA-75}^{pA-25} readcounts_j$$

$$N_i^{down} = 1/50 \sum_{j=pA+25}^{pA+75} readcounts_j$$

Transcriptome wide averages of $N^{up}$ and $N^{down}$ are defined as

$$N_A^{up} = 1/|A| \sum_i^{|A|} N_i^{up}$$

$$N_A^{down} = 1/|A| \sum_i^{|A|} N_i^{down}$$

Finally the processing index is given by

$$PI = log2\left( \frac{N_A^{down}}{max\left(1, \left(N_A^{up} - N_A^{down}\right)\right)} \right)$$

Colocalization analysis (**Figure 3C**) was done as described (**Baejen et al., 2014**), with modifications. Briefly, to calculate the tendency of pairs of factors A and B to bind locations in the transcriptome near each other, we computed the average occupancy of factor B within ±20 nt of occupancy peaks of factor A (unsmoothed occupancy data). First, crosslink sites of factor A are sorted according to their occupancy and the strongest n = 3000 sites are selected. For each crosslink site $a_i$ of this selection the maximum occupancy value of factor B $m_i^B$ is identified based on the occupancies of factor B 20 nt $\pm$ around $a_i$. The average colocalization $c$ is then given by $1/n \sum_i^n m_i^B$. Next, the background binding $b$ of factor B is defined as the median of all occupancies of factor B. The colocalization is defined as $log2(c/b)$. Finally, we constructed a data matrix containing the calculated colocalization values between all EF pairs. After data normalization the derived colocalization dissimilarity matrix (Euclidean distance) was subjected to average-linkage hierarchical clustering (**Figure 3C**).

## ChIP-Seq

ChIP was performed as described (**Mayer et al., 2010**), with modifications. Yeast strains were grown in 600 mL YPD medium to mid-log phase (OD600, ~0.8). Cell cultures were treated with formaldehyde (1%, Sigma, F1635) for 20 min at 20°C. Crosslinking was quenched with 75 mL of 3 M glycine for 5 min at 20°C. All subsequent steps were performed at 4°C with pre-cooled buffers and in the presence of a fresh protease-inhibitor mix (1 mM Leupetin, 2 mM Pepstatin A, 100 mM Phenylmethylsulfonyl fluoride, 280 mM Benzamidine). Cells were collected by centrifugation, washed with 1 × TBS (20 mM Tris-HCL at pH 7.5, 150 mM NaCl) and twice with lysis buffer (50 mM HEPES-KOH at pH 7.5, 150 mM NaCl, 1 mM EDTA, 1% Triton X-100, 0.1% Na deoxycholate, 0.1% SDS). Cell pellets were resuspended in 2 mL lysis buffer and cell lysis performed as described above for PAR-CLIP.

Chromatin was washed with lysis buffer and solubilized via sonication with a Covaris S220 instrument (Covaris) to yield an average DNA fragment size of 200 bp as determined on an Agilent 2200 TapeStation instrument (see below). This was achieved by sonicating the sample for 18 min using the following parameters: Peak Incident Power (W): 140; Duty Factor: 5%; Cycles per Burst: 200. 30

µL of the washed and fragmented chromatin samples were saved as input material and for control of the average chromatin fragment size. The remaining chromatin sample was immunoprecipitated with 100 µL antibody-coated and prewashed magnetic Dynabeads Protein G (Life Technologies, UK ) at 4°C for 3 hr (ChIP of TAP-tagged proteins) or overnight (ChIP with protein-specific antibodies) on a turning wheel. Immunoprecipitated chromatin was washed five times with ChIP wash buffer (100 mM Tris-HCl at pH 7.5, 500 mM LiCl, 1% NP-40, 1% Na deoxycholate) and one time with TE buffer (10 mM Tris-HCl at pH 7.5, 1 mM EDTA). Immunoprecipitated chromatin was eluted for 10 min at 70°C in the presence of ChIP elution buffer (100 mM Sodium bicarbonate, 1% SDS). Eluted immunoprecipitated chromatin as well as input material were incubated with 10 µL RNase A (10 mg/mL) at 37°C for 30 min and subsequently subjected to Proteinase K (20 µL of 20 mg/mL Proteinase K, Bioline, BIO-37084, Germany) digestion at 37°C for 2 hr and reversal of crosslinks (at 65°C overnight).

IP DNA and input samples were purified with the QIAquick MinElute PCR Purification Kit (Qiagen, Germany) according to the manufacturer's instructions. Elution was performed adding three times 15 µL H$_2$Odd to the columns with a 5 min incubation time in between. The average chromatin fragment size for each experiment was verified using 1 µL of the purified input samples on an Agilent 2200 TapeStation instrument using a D1000 ScreenTape (Agilent Technologies). DNA concentration of the IP and Input samples was determined with Qubit 1.0, dsDNA HS (Invitrogen). Before preparing libraries for Illumina sequencing, IP and input samples were analyzed via qRT-PCR on four housekeeping genes to assess quality of sample DNA (see below). For Illumina sequencing of ChIP samples, 1–10 ng of IP or Input DNA were used for library preparation according to the manufacturer's recommendations using the ThruPLEX DNA-Seq Kit (Rubicon Genomics, Inc., Ann Arbor, MI). Libraries were qualified on an Agilent 2200 TapeStation instrument and quantified with Qubit 1.0. Libraries were pooled and sequenced on an Illumina HiSeq 1500 sequencer.

## Quantitative real-time PCR

For ChIP experiments, input and immunoprecipitated (IP) samples were analyzed by qPCR to assess the extent of protein occupancy at different genomic regions. Primer pairs directed against promoter, coding and terminator regions of the housekeeping genes *PMA1*, *PDC1*, *MUP1*, *ILV5*, *ADH1* and *ALD5* as well as against a heterochromatic control region of chromosome V (chrV) were designed and the corresponding PCR efficiencies determined. All primer pairs used in this study had PCR efficiencies in the range of 95-100%. PCR reactions contained 1 µL DNA template, 1.6 µL of 10 µM primer pairs and 10 µL iQ SYBR Green Supermix (Bio-Rad, Hercules, CA). Quantitative PCR was performed on a qTOWER 2.2 Real-Time System (Analytik Jena AG, Germany) using a 3 min denaturing step at 95°C, followed by 40 cycles of 15 s at 95°C, 30 s at 61°C and 15 s at 72°C. Threshold cycle (Ct) values were determined using the corresponding qPCRsoft 3.1 software. Percent input was determined for each IP sample: 100*2^(Adjusted input - Ct (IP); Adjusted input: Raw Ct Input - log2 of Adjusted input to 100%. Sequence information of primer pairs is available in *Supplementary file 1*.

## ChIP-Seq data processing and analysis

Paired-end 50 bp reads were aligned to the *S. cerevisiae* genome (sacCer3, version 64.2.1) using the short read aligner Bowtie (version 2.2.3) (*Langmead and Salzberg, 2012*). SAMTools was used to quality filter SAM files (*Li et al., 2009*). Alignments with MAPQ smaller than 7 (-q 7) were skipped and only proper pairs (-f99, -f147, -f83, -f163) were selected. The BEDTools toolset (*Quinlan and Hall, 2010*) was used to obtain coverage tracks that were subsequently imported into R/Bioconductor where further processing of the data was carried out. Normalization between IP and Input was done using the signal extraction scaling (SES) factor obtained with the estimateScaleFactor function from deepTools (*Ramírez et al., 2014*) with options: –l 100 –n 100000 and the median fragment size (-f) estimated from the data (around 200 bp). ChIP enrichments were obtained by dividing SES-normalized IP intensities by the corresponding input intensities: log2(IP/Input).

The same transcript annotations as for PAR-CLIP data analysis (see above) were used for ChIP-Seq data analysis, except that filtering criteria had to be more stringent due to the lack of strand-specificity and lower resolution of ChIP-Seq data. Thus, for *Figure 7*, the distance filtering between transcripts was increased to 200 bp and transcripts on both strands were considered.

## Chromatin association assay

Yeast cultures were grown in 200 mL of YPD medium at 30°C to mid-log phase (OD$_{600}$, 0.8). Subsequent steps were performed at 4°C with precooled buffers and in the presence of a fresh protease-inhibitor mix. Cells were collected by centrifugation, washed with 1× TBS buffer and with lysis buffer (50 mM Tris-HCl pH 7.5, 100 mM NaCl, 0.5% sodium deoxycholate, 0.1% SDS, 0.5% NP-40). Cell pellets were flash frozen in liquid nitrogen and stored at −80°C. Pellets were thawed, resuspended in 1 mL lysis buffer, and disrupted via beat beating (see PAR-CLIP Materials and methods). The lysate was divided into two samples. One half was treated with 7.5 U of RNase A and 300 U of RNase T1 (Ambion, UK); the other half was treated with the same volume of the RNase storage buffer (10 mM HEPES pH 7.5, 20 mM NaCl, 0.1% Triton X-100, 1 mM EDTA, 50% glycerol). After 30 min incubation at room temperature, chromatin was isolated by centrifugation at 15,000 rpm for 15 min. Chromatin was solubilized in 1 mL lysis buffer via sonication with a Covaris S220 instrument (COVARIS, INC.). Chromatin solutions were then analyzed by SDS-PAGE and Western blotting against the C-terminal TAP tag of the analyzed factor and against H3, which served as loading control. We performed three independent biological replicates for TFIIB, Rpb1, Rtf1, Paf1, Ctk2, Bur1, Set2 and Spt6 and two independent biological replicates for Leo1, Ctk1, Bur1, Set1, Dot1 and Spt5. Band intensities were quantified using ImageJ software (1.49v). For statistical analysis, multiple group comparisons were done by one-way ANOVA with Dunnett post-hoc test. Data are presented as mean ± SD. Differences were considered significant when $p < 0.05$ (*$p < 0.05$; **$p < 0.01$; ***$p < 0.001$; n.s. = not significant).

## Cloning and expression of *S. cerevisiae* CTDK-I protein complex

The full-length subunits of the CTDK-I complex, Ctk1, Ctk2 and Ctk3 were amplified from genomic yeast DNA and cloned into modified pFastBac vectors via ligation independent cloning (LIC) (a gift of Scott Gradia, UC Berkeley, vectors 438-A, 438 C (Addgene: 55218, 55220)). Ctk2 bears an N-terminal 6x His MBP tag followed by a tobacco etch virus (TEV) protease cleavage site. Individual subunits were combined into a single plasmid by successive rounds of ligation independent cloning. Each subunit is preceded by a PolH promoter and followed by an SV40 termination site. Purified plasmid DNA (0.5 µg) was electroporated into DH10EMBacY cells to generate bacmids (*Berger et al., 2004*). Bacmids were prepared from positive clones by isopropanol precipitation and transfected into Sf9 cells (ThermoFisher, UK) grown in Sf-900 III SFM (ThermoFisher) with X-tremeGENE9 transfection reagent (Sigma) to generate V0 virus. V0 virus was harvested 72 hr after transfection. V1 virus was produced by infecting 25 mL of Sf21 cells (Expression Technologies, UK) grown at 27°C, 300 rpm with V0 virus (1E6 cell/mL, 1:50 (v/v) cells:virus). V1 viruses were harvested 48 hr after proliferation arrest and stored at 4°C. For protein expression, 600 mL of Hi5 cells (Expression Technologies) (1E6/mL) grown in ESF921 medium (Expression Technologies) were infected with 200 µL of V1 virus and grown for 72 hr at 27°C. Cells were harvested by centrifugation (238x g, 4°C, 30 min), resuspended in lysis buffer at 4°C (400 mM NaCl, 20 mM Na•HEPES pH 7.4, 10% glycerol (v/v), 1 mM DTT, 30 mM imidazole pH 8.0, 0.284 µg/mL leupeptin, 1.37 µg/mL pepstatin A, 0.17 mg/mL PMSF, 0.33 mg/mL benzamidine), snap frozen, and stored at −80°C. All insect cell lines were not tested for mycoplasma contamination and the identity of the cells was not confirmed.

## Purification of complete CTDK-I kinase complex

Protein purification steps were performed at 4°C. Frozen cell pellets were thawed and lysed by sonication. Lysates were clarified by centrifugation in an A27 rotor (ThermoFisher) (26,195 xg, 4°C, 30 min), followed by ultracentrifugation in a Type 45 Ti rotor (Beckman Coulter, Brea, CA) (235,000 xg, 4°C, 60 min). Clarified lysates were filtered through 0.8 mm syringe filters (Millipore, Billerica, MA) and applied to a 5 mL HisTrap column (GE Healthcare, UK) equilibrated in lysis buffer. HisTrap columns were washed with 10CV of lysis buffer followed by 5CV of high salt wash buffer (800 mM NaCl, 20 mM Na•HEPES pH 7.4, 10% glycerol (v/v), 1 mM DTT, 30 mM imidazole pH 8.0, 0.284 µg/ml leupeptin, 1.37 µg/ml pepstatin A, 0.17 mg/ml PMSF, 0.33 mg/ml benzamidine) and 5CV of lysis buffer. An amylose column (NEB) equilibrated in lysis buffer was directly coupled to the HisTrap column. Protein was eluted from the HisTrap column by a gradient from 0–100% nickel elution buffer (400 mM NaCl, 20 mM Na•HEPES pH 7.4, 10% glycerol (v/v), 1 mM DTT, 500 mM imidazole pH 8.0, 0.284 µg/mL leupeptin, 1.37 µg/mL pepstatin A, 0.17 mg/mL PMSF, 0.33 mg/ml benzamidine), after

which the HisTrap and amylose column were decoupled. The amylose column was washed with 5CV of lysis buffer and protein was eluted with amylose elution buffer (400 mM NaCl, 20 mM Na•HEPES pH 7.4, 10% glycerol (v/v), 1 mM DTT, 30 mM imidazole pH 8.0, 10 g maltose, 0.284 µg/mL leupeptin, 1.37 µg/mL pepstatin A, 0.17 mg/mL PMSF, 0.33 mg/ml benzamidine). Peak fractions were analyzed by SDS-PAGE. Amylose column fractions containing CTDK-I were combined with 1.5 mg of His6-TEV protease and dialyzed overnight at 4°C in a Slide-A-Lyzer (3–12 ml 10 kDa MWCO) (ThermoFisher) against 1 L of lysis buffer. Protein was removed from the Slide-A-Lyzer cassette and applied to a 5 mL HisTrap column to remove uncleaved protein and TEV protease. Protein was concentrated in an Amicon 15 mL centrifugal 30 MWCO concentrator (Millipore) to 500 µL. The protein was applied to a Superdex 200 Increase 10/300 column (GE Healthcare) equilibrated in 400 mM NaCl, 20 mM Na•HEPES pH 7.4, 10% (v/v) glycerol, and 1 mM DTT. Peak fractions were analyzed by SDS-PAGE. Pure fractions were concentrated as described above to 500 µL, aliquoted, flash frozen, and stored at −80°C. Protein preparation yielded ~4 mg of full-length CTDK-I from 3.8 L of insect cell culture.

## In vitro CTDK-I kinase activity assays

Purified human GST-CTD (see below) and *S. cerevisiae* Pol II (*Plaschka et al., 2016*) were used as kinase substrates. In vitro kinase assays were performed in 20 µL reactions. Recombinant purified CTDK-I (0.4 µM) was mixed together with either GST-CTD (10 µM) or Pol II (2 µM) and the following final conditions: 1 mM DTT, 100 mM NaCl, 30 mM Na•HEPES pH 7.5, 4% (v/v) glycerol and 3 mM $MgCl_2$. Reactions were incubated at 30°C for 5 min; 1 µL '0 min' time point was taken before adding 3 mM ATP. Reactions were then incubated at 30°C with constant shaking at 300 rpm. 2 µL were taken for each time point (reaction with CTD substrate: 5, 10, 20, 30, 60 and 120 min; reaction with Pol II substrate: 2, 4, 6, 10, 20, 30, 60 and 90 min). The reactions were stopped with LDS-loading buffer and heated at 95°C for 5 min before electrophoresis on 4–12% SDS polyacrylamide gel. Standard methods were used for Western blotting. 1:14 and 1:3000 dilutions were used for anti-Ser2-P primary antibody (3E10) (*Chapman et al., 2007*) and anti-rat-HRP secondary antibody (Sigma; A9037), respectively. SuperSignal West Pico Chemiluminescent Substrate (ThermoFischer) was used for detection.

## Fluorescence anisotropy assays with CTDK-I

5′-FAM labeled ssRNA and dsDNA were obtained from Integrated DNA Technologies and dissolved in water to 100 µM. Sequences for ssRNA were 24% GC, A-rich (AAUAUUCAAGACGAUUUA-GACGAUAAUAUCAUA), 24% GC, U-rich (AUGUUGUAUGAUAUCUUGCUAACUUAAUUUGAU), 45% GC, A-rich (AAGCAGCCAAACAAGCAGUCAACAUCAAGUCGU) and 45% GC, U-rich (UUCG UCGGUUUGUGCGUCAGUUGUAGUUCAUCA). The dsDNA sequence corresponds to the 45% GC, A-rich ssRNA sequence. 24% GC, A-rich, 24% GC, U-rich and 45% GC, A-rich sequences correspond to natural coding sequences in *S. cerevisiae*. RNA oligos were unfolded by incubating the RNA at 95°C for 1 min and transferring to ice for 10 min. The oligonucleotide sequences were diluted in water for all experiments. Purified CTDK-I was serially diluted in two fold steps in dilution buffer (200 mM NaCl, 20 mM Na•HEPES pH 7.4, 1 mM DTT and 10% glycerol). Nucleic acids (8 nM final concentration) were added on ice and the reaction was incubated for 10 min. The assay was brought to a final volume of 30 µL and incubated for 20 min at RT in the dark (final conditions: 100 mM NaCl, 2 mM $MgCl_2$, 20 mM Na•HEPES pH 7.4, 1 mM TCEP, 4% glycerol, 0.01 mg/mL BSA and 5 µg/mL yeast tRNA (Sigma) as a competitor for non-specific binding). 18 µL of each reaction were transferred to a Greiner 384 Flat Bottom Black Small volume plate. Fluorescence anisotropy was measured at 30°C with an Infinite M1000Pro reader (Tecan, Switzerland) with an excitation wavelength of 470 nm (±5 nm), an emission wavelength of 518 nm (±20 nm) and a gain of 63. All experiments were done in triplicate and analyzed with GraphPad Prism Version 7. Binding curves were fitted with a single site quadratic binding equation:

$$y = \left( \frac{Bmax * \left( [x] + [L] + Kd,app - \sqrt{([x]+[L]+Kd,app)^2 - 4([x]*[L])} \right)}{2*[L]} \right)$$

where $B_{max}$ is the maximum specific binding, L is the concentration of nucleic acid, x is the concentration of CTDK-I, $K_{d,app}$ is the apparent disassociation constant for CTDK-I and nucleic acid. Error bars represent the standard deviation from the mean of three experimental replicates.

## His6-TEV-GST 52x human CTD expression and purification

For expression, Bl21(DE3)pLysS *E. coli* cells were used. Cells were grown in 2xYT to $OD_{600}$ of 0.5 (37°C, 160 rpm), changed to 18°C, and induced overnight with 0.5 mM IPTG. Protein purification steps were performed at 4°C. 4L Bl21(DE3)pLysS cells were thawed and lysed by sonication. Lysates were clarified by centrifugation in an A27 rotor (ThermoFisher) (26,195 xg, 4°C, 30 min). Clarified lysate was applied to a 5 mL HisTrap nickel column (GE Healthcare) equilibrated in lysis buffer (150 mM NaCl, 20 mM Na•HEPES pH 7.4, 1 mM DTT, 10% glycerol, 30 mM imidazole, 0.284 µg/ml leupeptin, 1.37 µg/ml pepstatin A, 0.17 mg/ml PMSF, 0.33 mg/ml benzamidine). The column was then washed in high salt wash buffer (800 mM NaCl, 20 mM Na•HEPEs pH 7.4, 1 mM DTT, 10% glycerol, 30 mM imidazole, 0.284 µg/ml leupeptin, 1.37 µg/ml pepstatin A, 0.17 mg/ml PMSF, 0.33 mg/ml benzamidine) followed by lysis buffer. A 5 mL HiTrap Q column was equilibrated in lysis buffer and added in line after the nickel column. Protein was eluted with a gradient of 0–100% nickel elution buffer (150 mM NaCl, 20 mM Na•HEPES pH 7.4, 1 mM DTT, 10% glycerol, 500 mM imidazole, 0.284 µg/ml leupeptin, 1.37 µg/ml pepstatin A, 0.17 mg/ml PMSF, 0.33 mg/ml benzamidine). Following imidazole elution, the nickel column was removed and the Q column washed in 150 mM NaCl, 20 mM Na•HEPES pH 7.4, 1 mM DTT, 10% glycerol, 0.284 µg/ml leupeptin, 1.37 µg/ml pepstatin A, 0.17 mg/ml PMSF, 0.33 mg/ml benzamidine. Gradient elution was performed with 1000 mM NaCl, 20 mM Na•HEPES pH 7.4, 1 mM DTT, 10% glycerol, 0.284 µg/ml leupeptin, 1.37 µg/ml pepstatin A, 0.17 mg/ml PMSF, 0.33 mg/ml benzamidine. Peak fractions from the Q column elution were concentrated in a 5 mL 30 kDa MWCO concentrator to 500 µL. Protein was centrifuged for 10 min at 15k prior loading to a Superdex 200 Increase 10/300 column (GE Healthcare), equilibrated in in SE buffer (300 mM NaCl, 20 mM Na•HEPES pH 7.4, 1 mM DTT, 10% glycerol). Peak fractions were analyzed by SDS-PAGE, pure fractions concentrated in 5 mL 30 MWCO Amicon Millipore concentrator to 40 µM, aliquoted, flash frozen, and stored at −80°C.

## Accession numbers

Data have been deposited in NCBI's Gene Expression Omnibus (*Edgar et al., 2002*) and are accessible through GEO Series accession number GSE81822.

## Acknowledgements

We thank Monika Raabe, Annika Kühn and Henning Urlaub for mass spectrometry analysis and A. Boltendahl for help with PAR-CLIP experiments. We also thank J. Söding for discussions concerning PAR-CLIP data evaluation and C. Roth for updating our PAR-CLIP pipeline. ML was funded by the Center for Innovative Medicine (CIMED) at Karolinska Institutet and by the Science for Life Laboratory (SciLifeLab). SMV was supported by an EMBO Long-Term Postdoctoral Fellowship (ALTF 745–2014). PC was supported by the Deutsche Forschungsgemeinschaft (SFB860, SPP1935) the European Research Council Advanced Investigator Grant TRANSREGULON (grant agreement No 693023), and the Volkswagen Foundation.

## Additional information

### Funding

| Funder | Grant reference number | Author |
| --- | --- | --- |
| European Molecular Biology Organization | ALTF 745-2014 | Seychelle M Vos |
| Center for Innovative Medicine | | Michael Lidschreiber |
| Science for Life Laboratory | | Michael Lidschreiber |
| Deutsche Forschungsgemeinschaft | SFB860 SPP1935 | Patrick Cramer |

| European Research Council | 693023 | Patrick Cramer |
|---|---|---|
| Volkswagen Foundation | | Patrick Cramer |
| Max Planck Institute for Bio-physical Chemistry | Open-access funding | Patrick Cramer |

The funders had no role in study design, data collection and interpretation, or the decision to submit the work for publication.

### Author contributions

SB, Conceptualization, Investigation, Visualization, Methodology, Writing—original draft, Writing—review and editing, Analysis and interpretation of data; ML, Conceptualization, Data curation, Investigation, Visualization, Writing—original draft, Writing—review and editing, Analysis and interpretation of data; CB, Methodology; PT, Analysis of data; SMV, Investigation, Methodology; PC, Conceptualization, Supervision, Funding acquisition, Writing—original draft, Project administration, Writing—review and editing

### Author ORCIDs

Sofia Battaglia, http://orcid.org/0000-0002-9861-5227
Michael Lidschreiber, http://orcid.org/0000-0002-6740-2755
Seychelle M Vos, http://orcid.org/0000-0003-1985-2994
Patrick Cramer, http://orcid.org/0000-0001-5454-7755

## Additional files

### Supplementary files

• Supplementary file 1. Sequences of primer pairs used for ChIP-qPCR. *YER*: Heterochromatic region on chromosome V. Sequences of primer pairs used for strain generation.

### Major datasets

The following dataset was generated:

| Author(s) | Year | Dataset title | Dataset URL | Database, license, and accessibility information |
|---|---|---|---|---|
| Battaglia S, Lidschreiber M, Baejen C, Torkler P, Vos S, Cramer P | 2017 | RNA contributes to chromatin association of transcription elongation factors and RNA polymerase II CTD kinases | https://www.ncbi.nlm.nih.gov/geo/query/acc.cgi?acc=GSE81822 | Publicly available at the NCBI Gene Expression Omnibus (accession no: GSE81822) |

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
