## [Decision Letter]

Thank you for submitting your article "RNA-dependent chromatin association of transcription elongation factors and Pol II CTD kinases" for consideration by *eLife*. Your article has been reviewed by two peer reviewers and a member of the Board of Reviewing Editors (BRE), and the evaluation has been overseen by the BRE and Dr. James Manley as the Senior Editor. The reviewers have opted to remain anonymous.

The reviewers and the BRE have discussed the reviews with one another and the Reviewing Editor has drafted this decision to help you prepare a revised submission.

Overall, the reviewers find the study very important and think that a revised version should be a great contribution to *eLife*. In addition to several more minor suggestions and textual improvements (included in the reviewers), the reviewers have requested two major additions to the manuscript:

1) The reviewers would like for you and colleagues to test the role of the RRM domain of Set1 in Set1 recruitment.

2) The reviewers also requested that you test the link between Paf1C subunits and Set1 association with chromatin/Pol II. It was previously demonstrated that Paf1C plays a role as a platform for Set1 association with elongating Pol II, can Set1 still associate with Pol II in the absence of Paf1C?

Also testing the role of CTD of Pol II Vs. RNA in Set2 recruitment to the elongating Pol II could be highly informative, however, not a requirement, but we think its addition will improve the manuscript.

*Reviewer #1:*

Overall this study is both informative and provides compelling evidence for the role of nascent transcription in the recruitment of EFs to genes. It remains a possibility that RNA:EF crosslinking occurs because of close proximity in the elongation complex. However, the RNAse treatment control (Figure 5) provides good evidence for a direct RNA role in EF recruitment at least for many of the tested EFs. Particularly compelling is the demonstrated specificity of EFs that have known effects at different stages of the transcription cycle yielding parallel RNA binding profiles.

Subsection “Comparisons of PAR-CLIP data require normalization”, first paragraph: Authors normalize data to RNA-Seq or Pol II PAR-CLIP. Is there any chromatin RNA-seq data available? Given that we are looking at nascent RNA this would be another interesting normalization method. Another possibility is normalization to NET-Seq signal (Churchman and Weissman 2010).

Subsection “Comparisons of PAR-CLIP data require normalization”, second paragraph: The authors point out that the 3' part of the transcripts has higher RNA-Seq signal. Is this generally expected? Is there an explanation for this phenomenon?

Subsection “Comparisons of PAR-CLIP data require normalization”, last paragraph: It would be nice to see some unstable profiles with the three normalization methods. Or is there no RNA-Seq signal at all for unstable transcripts? Again, Chr RNA-Seq or NET-Seq would then be a nice alternative.

Figure 3—figure supplement 1. Some EFs seem to bind more to exons than to introns (e.g. Set2, Dot1) – wouldn't this suggest that there is a lot of mature RNA? On the other hand, some introns show more EF binding than exons (e.g. Set1, Spt5). How does this happen?

Subsection “Most EFs preferentially interact with coding transcripts”, first paragraph: How well defined is the TTS for unstable transcripts?

Figure 1 legend: "Non-averaged heat plots" – the heat plots are still averaged across mRNA transcripts. Do the authors mean non-position averaged? Or do they mean Figure 3—figure supplement 1?

Figure 4 and the corresponding supplement have no legend.

Subsection “PAR-CLIP data analysis”, second paragraph: What does window half size mean? Does that mean the moving window is 60nt, 30nt to either side of the position? Please clarify.

Subsection 2 PAR-CLIP data analysis”, second paragraph 100% occupancy is a misleading terminology.

Figure 1—figure supplement 1 legend: Correlation need to be quantified. Would it be possible to present some scatter plots with the PAR-CLIP signals at mRNAs and the corresponding correlations?

Subsection “PAR-CLIP data analysis”, last paragraph: The authors do not define "b", which I suspect is "background binding".

Subsection “Chromatin association of EFs depends on RNA”, end of first paragraph: The authors speak of "discrepancies" between chromatin association and PAR-CLIP. It is not clear what these discrepancies exactly are. Spt6 is not in the PAR-CLIP data, Leo1 and Paf1 had little PAR-CLIP signal and would not be expected to be affected by Rnase – so what are the discrepancies?

Figure 2: Shades of legend do not correspond to the graph. Given the rest of the figure is in colour, could this one also be in colour? Otherwise please adjust the shading.

Figures and supplemental figures are not labelled.

*Reviewer #2:*

This manuscript from the Cramer lab represents an interesting study that documents the ability of many transcription elongation factors and histone modifiers to associate with nascent and stable RNA during RNA Pol II transcription in yeast. They show using PAR-CLIP that all elongation factors and modifiers that they tested can crosslink to RNA, and further, that this RNA binding is largely similar to their localization patterns along genes. Among a number of important takeaways, the authors show that the association of several elongation factors to chromatin is dependent on RNA interaction in vitro, thus providing a potential mechanism for how these factors are recruited to genes and RNA Pol II in vivo.

This study is sure to be on interest to the broad chromatin and transcription community. One recommendation to improve this study, however, is to include an experiment that would validate the role of RNA binding on in vivo targeting and function on one of these elongation factors, which is currently lacking. For example, Set1 has several RRM motifs that would be ripe to mutate and test in this assay. Would blocking RNA binding prevent Set1 recruitment to genes? I think including an experiment that addresses this obvious and unresolved question is warranted.

---

## [Author Response]

*Overall, the reviewers find the study very important and think that a revised version should be a great contribution to eLife. In addition to several more minor suggestions and textual improvements (included in the reviewers), the reviewers have requested two major additions to the manuscript:*

*1) The reviewers would like for you and colleagues to test the role of the RRM domain of Set1 in Set1 recruitment.*

This is a very good suggestion. We generated a *S. cerevisiae* strain lacking the N-terminal region of Set1 that contains both RRMs; we also introduced a C-terminal TAP-tag on the mutated Set1 for protein detection and immunoprecipitation. We carried out Set1 ChIP-qPCR at four different housekeeping genes and compared gene occupancies between WT and mutant strains. Indeed, the RRM domains of Set1 turned out to be important for gene recruitment of the protein, supporting our model, in which the nascent RNA contributes to the recruitment of elongation factors to chromatin. We have included these interesting results that we feel strengthen the paper.

*2) The reviewers also requested that you test the link between Paf1C subunits and Set1 association with chromatin/Pol II. It was previously demonstrated that Paf1C plays a role as a platform for Set1 association with elongating Pol II, can Set1 still associate with Pol II in the absence of Paf1C?*

This is another good suggestion. We deleted the Paf1 gene in a Set1-TAP strain. As above, we carried out Set1 ChIP-qPCR at four different housekeeping genes and compared gene occupancies between WT and mutant strains. As expected and reported in the literature, deletion of Paf1 leads to a decrease in Set1 gene occupancy, providing evidence for Paf1C contributing to recruitment. Taken together, our ChIP-qPCR results suggest that both Paf1C and the RRM domains of Set1 are important for full Set1 recruitment to genes. We present these data in a new figure (Figure 8).

*Also testing the role of CTD of Pol II Vs. RNA in Set2 recruitment to the elongating Pol II could be highly informative, however, not a requirement, but we think its addition will improve the manuscript.*

We agree that this experiment will be interesting to further investigate the recruitment of Set2. However, since the RNA-binding region of Set2 is currently not known we were not able to do the suggested experiment. We will, however, keep this in mind for future studies.

*Reviewer #1:*

*[…] Subsection “Comparisons of PAR-CLIP data require normalization”, first paragraph: Authors normalize data to RNA-Seq or Pol II PAR-CLIP. Is there any chromatin RNA-seq data available? Given that we are looking at nascent RNA this would be another interesting normalization method. Another possibility is normalization to NET-Seq signal (Churchman and Weissman 2010).*

Unfortunately, we are not aware of any chromatin RNA-Seq data available for *S. cerevisiae*, but we agree with the reviewer that using the yeast NET-Seq data from the Weissman lab represents an interesting alternative to use for normalization. Indeed, we used this data for normalization trials before we generated our Pol II PAR-CLIP data. Similar to our Pol II PAR-CLIP data, NET-Seq data detects unstable transcripts better than total RNA-Seq, and thus NET-Seq normalized and Pol II PAR-CLIP normalized profiles were generally similar. Nevertheless, we wish to keep our Pol II PAR-CLIP data as a reference, since these data were obtained using identical conditions. NET-Seq data was obtained using a different protocol and different library preparation strategy, which might lead to unwanted biases.

*Subsection “Comparisons of PAR-CLIP data require normalization”, second paragraph: The authors point out that the 3' part of the transcripts has higher RNA-Seq signal. Is this generally expected? Is there an explanation for this phenomenon?*

Since we are looking at meta-profiles of total RNA data, which generally show reduced levels of intronic reads, the average signal is expected to be a bit lower in the 5' part of genes since most introns in *S. cerevisiae* are located at the beginning of genes. Moreover, library preparation can introduce biases leading to higher signals towards the 3' end of genes. The latter point highlights the importance of using a reference control sample for normalization obtained using the same protocols and same library preparation strategy.

*Subsection “Comparisons of PAR-CLIP data require normalization”, last paragraph: It would be nice to see some unstable profiles with the three normalization methods. Or is there no RNA-Seq signal at all for unstable transcripts? Again, Chr RNA-Seq or NET-Seq would then be a nice alternative.*

We thank the reviewer for the comment. We now compare meta-profiles over unstable CUTs versus stable mRNAs using all three normalization methods. We added this as Figure 2—figure supplement 1. This comparison shows nicely that for proteins that bind CUTs (e.g. Spt5) the signal over unstable CUTs is artificially increased when total RNA-Seq reads are used for normalization, as was seen for unstable transcripts downstream of the pA site. Due to the reasons described in the previous points, we would prefer to stick to the Pol II PAR-CLIP data for normalization. Indeed, in the new Figure 2—figure supplement 1, we show that after Pol II PAR-CLIP normalization, Spt5 signals are equally high at mRNAs and CUTs, while no RNA normalization leads to less Spt5 signal at CUTs (due to less transcription of CUTs) and total RNA normalization leads to increased Spt5 levels at CUTs (due to decreased detection of unstable transcripts). Since Spt5 binds Pol II immediately after initiation, differences in Spt5 RNA binding between mRNAs and CUTs would not be expected, arguing that Pol II normalization helps to correctly interpret the data.

*Figure 3—figure supplement 1. Some EFs seem to bind more to exons than to introns (e.g. Set2, Dot1) – wouldn't this suggest that there is a lot of mature RNA? On the other hand, some introns show more EF binding than exons (e.g. Set1, Spt5). How does this happen?*

Most introns in *S. cerevisiae* are located at the beginning of transcripts and splicing occurs very fast (see Oesterreich/Neugebauer Cell 2016). Thus, we would expect slight differences in exon versus intron binding of EFs depending on when these EFs are recruited to elongating Pol II. Set2 and Dot1 are recruited during transcription elongation, later than Spt5 and Set1. Thus, many introns may already be spliced by the time that Set2/Dot1 are recruited, leading to less intron signal. Spt5 and Set1 are recruited immediately after transcription initiation and, therefore, one would expect Pol II normalized signals to be equal over introns and exons. Slightly increased intron signal for Spt5/Set1 may be explained by a longer residence time of these factors on introns than Pol II itself, but for reasons that are presently unknown.

*Subsection “Most EFs preferentially interact with coding transcripts”, first paragraph: How well defined is the TTS for unstable transcripts?*

In general TTSs for unstable transcripts like CUTs can be quite heterogenous. The CUT TTSs used here represent the most prominent TTSs, i.e. the point where most of the RNA signal drops. These CUT TTSs were identified by Xu et al. using a segmentation algorithm and subsequent manual curation. They used a deletion mutant for the nuclear exosome subunit RRP6 to stabilize CUTs and make their detection easier.

*Figure 1 legend: "Non-averaged heat plots" – the heat plots are still averaged across mRNA transcripts. Do the authors mean non-position averaged? Or do they mean Figure 3—figure supplement 1?*

We thank the reviewer for this observation. The reviewer is right, the term “Non-averaged heat plots” is misleading. Whereas the PAR-CLIP signal in Figure 1 is averaged two times (i.e. mean transcript PAR-CLIP signals averaged over all mRNAs) the heat plots in Figure 1—figure supplement 1 are also averaged, but only once (i.e. per position over all mRNAs). We have clarified this in the legend.

*Figure 4 and the corresponding supplement have no legend.*

We thank the reviewer for this observation. We added the appropriate figure legend in the revised article.

*Subsection “PAR-CLIP data analysis”, second paragraph: What does window half size mean? Does that mean the moving window is 60nt, 30nt to either side of the position? Please clarify.*

What we mean by window half size 30 is that the window is 61 nt, 30 nt to either side of the current position. We are sorry if this was not clear. We have described it more clearly in the revised Methods.

*Subsection 2 PAR-CLIP data analysis”, second paragraph 100% occupancy is a misleading terminology.*

We agree. We have changed “scaled between 0 (0% occupancy) and 1 (100% occupancy)” to “scaled between 0 (minimum signal) and 1 (maximum signal)”.

*Figure 1—figure supplement 1 legend: Correlation need to be quantified. Would it be possible to present some scatter plots with the PAR-CLIP signals at mRNAs and the corresponding correlations?*

We agree. In addition to the average mRNA PAR-CLIP profiles shown for biological replicates in Figure 1—figure supplement 1, we now also show scatter plots with the PAR-CLIP signals at mRNAs (#T>C transitions per mRNA) and the corresponding Pearson correlations. Correlations between biological replicates were generally high, ranging between 0.94 and 0.99.

*Subsection “PAR-CLIP data analysis”, last paragraph: The authors do not define "b", which I suspect is "background binding".*

We thank the reviewer for pointing this out. It is correct that background binding is defined by b. It is now correctly defined in the revised Methods.

*Subsection “Chromatin association of EFs depends on RNA”, end of first paragraph: The authors speak of "discrepancies" between chromatin association and PAR-CLIP. It is not clear what these discrepancies exactly are. Spt6 is not in the PAR-CLIP data, Leo1 and Paf1 had little PAR-CLIP signal and would not be expected to be affected by Rnase – so what are the discrepancies?*

Generally, the presented chromatin association and PAR-CLIP data are consistent. The few discrepancies we wanted to refer to were Dot1 (low PAR-CLIP signal versus strong RNA-dependent chromatin association) and Spt6 (high PAR-CLIP signal versus RNA-independent chromatin association). We changed the text in the revised article to make this point clear.

*Figure 2: Shades of legend do not correspond to the graph. Given the rest of the figure is in colour, could this one also be in colour? Otherwise please adjust the shading.*

We adjusted the color and legend of Figure 2.

*Figures and supplemental figures are not labelled.*

We fixed this in the revised article.

*Reviewer #2:*

*[…] This study is sure to be on interest to the broad chromatin and transcription community. One recommendation to improve this study, however, is to include an experiment that would validate the role of RNA binding on* in vivo *targeting and function on one of these elongation factors, which is currently lacking. For example, Set1 has several RRM motifs that would be ripe to mutate and test in this assay. Would blocking RNA binding prevent Set1 recruitment to genes? I think including an experiment that addresses this obvious and unresolved question is warranted.*

We thank the reviewer for this recommendation. As described above, we generated a *S. cerevisiae* strain lacking the N-terminal region of Set1 that contains both RRMs; we also introduced a C-terminal TAP-tag on the mutated Set1 for protein detection and immunoprecipitation. We carried out Set1 ChIP-qPCR at 4 different housekeeping genes and compared gene occupancies between WT and mutant strains. Indeed, the RRM domains of Set1 are important for gene recruitment of the protein, supporting our model, in which the nascent RNA helps to recruit elongation factors to chromatin. We present these data in a new Figure (Figure 8).